# Accelerated water activation and stabilized metal-organic framework via constructing triangular active-regions for ampere-level current density hydrogen production

Fanpeng Cheng[1], Xianyun Peng[2], Lingzi Hu[3], Bin Yang[1,2], Zhongjian Li[1,2], Chung-Li Dong [4], Jeng-Lung Chen [5], Liang-Ching Hsu[5], Lecheng Lei [1,2], Qiang Zheng[6], Ming Qiu[3] ✉, Liming Dai[7] ✉ & Yang Hou [1,2,8,9] ✉

Two-dimensional metal-organic frameworks (MOFs) have been explored as effective electrocatalysts for hydrogen evolution reaction (HER). However, the sluggish water activation kinetics and structural instability under ultrahigh-current density hinder their large-scale industrial applications. Herein, we develop a universal ligand regulation strategy to build well-aligned Ni-benzenedicarboxylic acid (BDC)-based MOF nanosheet arrays with S introducing (S-NiBDC). Benefiting from the closer $p$-band center to the Fermi level with strong electron transferability, S-NiBDC array exhibits a low overpotential of 310 mV to attain $1.0\ A\ cm^{-2}$ with high stability in alkaline electrolyte. We speculate the newly-constructed triangular "$Ni_2$-$S_1$" motif as the improved HER active region based on detailed mechanism analysis and structural characterization, and the enhanced covalency of Ni-O bonds by S introducing stabilizes S-NiBDC structure. Experimental observations and theoretical calculations elucidate that such Ni sites in "$Ni_2$-$S_1$" center distinctly accelerate the water activation kinetics, while the S site readily captures the H atom as the optimal HER active site, boosting the whole HER activity.

High-purity hydrogen production through alkaline electrochemical water splitting is widely considered as one of the cleanest and most promising processes[1–3]. Expensive and scarce noble metal (e.g., Pt)-based materials are still considered the best electrocatalysts for the hydrogen evolution reaction (HER) involved in water electrolysis[4]. Despite considerable efforts that have been made to explore cost-effective Pt alternatives, their unsatisfied activity and stability,

especially at the industrial current densities of $>1.0\ A\ cm^{-2}$, remain as the major barrier for vital applications[5,6].

Recently, metal-organic frameworks (MOFs), constructed from well-dispersed metal/cluster nodes periodically coordinated with organic linkers of a tunable porous structure and defined crystal structure, have been widely investigated as candidate catalysts for HER[7–9]. To improve their catalytic stability and conductivity, MOF

[1]Key Laboratory of Biomass Chemical Engineering of Ministry of Education, College of Chemical and Biological Engineering, Zhejiang University, Hangzhou 310027, China. [2]Institute of Zhejiang University - Quzhou, Quzhou 324000, China. [3]Institute of Nanoscience and Nanotechnology, College of Physical Science and Technology, Central China Normal University, Wuhan 430079, China. [4]Department of Physics, Tamkang University, Tamsui 25137, Taiwan. [5]National Synchrotron Radiation Research Center, Hsinchu 30076, Taiwan. [6]CAS Key Laboratory of Standardization and Measurement for Nanotechnology, CAS Center for Excellence in Nanoscience, National Center for Nanoscience and Technology, Beijing 100190, China. [7]Australian Carbon Materials Centre (A-CMC), School of Chemical Engineering, University of New South Wales, Sydney, NSW 2052, Australia. [8]School of Biological and Chemical Engineering, NingboTech University, Ningbo 315100, China. [9]Donghai Laboratory, Zhoushan, China. ✉e-mail: qium@mail.ccnu.edu.cn; l.dai@unsw.edu.au; yhou@zju.edu.cn

materials are usually used as sacrificial templates, from which carbon-based electrocatalysts are derived by heating treatment[10,11]. However, high-temperature calcination often causes severe structure destruction, metal agglomeration, and considerably low utilization of the MOF templates[12]. Therefore, it is highly desirable to rationally design nanostructured MOFs with accessible active sites as electrocatalysts directly without further heat treatment.

For electrocatalytic HER processes in alkaline, it typically follows a Volmer-Heyrovsky mechanism[13,14], involving: (i) adsorption and dissociation of $H_2O$ molecule via water activation to form H* intermediate (Volmer step): $H_2O^* + e^- \rightarrow H^* + OH^-$; and (ii) combination of the adsorbed H* with an electron transferred from electrode and one $H_2O$ molecule to form one $H_2$ molecule (Heyrovsky step): $H^* + H_2O + e^- \rightarrow * + H_2 + OH^-$. For MOFs-based catalysts with saturated symmetrically coordinated metal nodes that are unfavorable to adsorb and dissociate water, however, the sluggish water activation kinetics in the Volmer step is the main stumbling block for them to exhibit good HER activities[15–17]. Thereby, it is important to break the spatial symmetrical structure of metal centers in MOFs to accelerate water activation, thus improving the overall HER activity[18,19]. Inspired by [FeFe]-hydrogenase, an enzyme in algae containing special "$Fe_2$-$S_1$" active units and the fastest $H_2$ production catalyst known in nature[20,21], we developed in this study a unique active region consisting of dual-metal atoms bridged with S atom in MOFs to form asymmetrical configurations for facilitating water activation and enhancing their alkaline HER performance. We further found that manipulating the center metal configurations is also an effective way to improve the stability of MOFs for HER electrocatalysis. This possibility has not been recognized, although the activity of currently-reported MOF catalysts decreases rapidly at high current densities necessary for industrial electrolyzers due to the active phase transformation and element dissolution[22,23].

Herein, we report a universal ligand modulation strategy to synthesize a S-doped Ni-benzenedicarboxylic acid (BDC)-based MOF (S-NiBDC)[24]. The newly-developed S-NiBDC nanosheet arrays grew on a conductive Ni foam with a thickness of ~16 nm and S-dopant content of 1.67 wt.%. Owing to an upshift of the p-band center and fast electron transfer, S-NiBDC array exhibits a low overpotential of 310 mV to reach 1.0 A cm$^{-2}$ and a small Tafel slope of 75 mV dec$^{-1}$ towards alkaline HER with superb stability, superior to almost all previously-reported MOF-based HER electrocatalysts at high ampere-level current densities and even better than the state-of-the-art Pt/C (462 mV at 1.0 A cm$^{-2}$). Detailed structural investigations revealed that the coordination of S species with Ni sites broke the pristine symmetrical metal sites, which might result in a special triple-atom "$Ni_2$-$S_1$" active center (Fig. 1) to enhance the HER activity. Besides, the S modulation stabilizes the Ni-O bonds to prevent the MOFs from possible structure damage, and hence improved electrocatalytic stability. Overall, in the ternary "$Ni_2$-$S_1$" active region, the S-modified Ni sites can efficiently trigger water activation kinetics to promote the rate-determining Volmer step, and the S site can further trap the H atom to form S-H* intermediate for facilitating Heyrovsky step ($H_2$ evolution), eventually accelerating the alkaline HER process.

## Results
### Theory predictions for density of states
To predict the competence of constructing triangular active regions in BDC-based MOFs by a ligand modulation strategy for promoting HER activities, we firstly performed density functional theory (DFT) calculations. As shown in Fig. 1a, b, two layered-pillared NiBDC and S-NiBDC models are constructed, in which the Ni atoms are coordinated with BDC organic linkers to form NiBDC structure, while partial BDC ligands are replaced by 1,4-benzenedimethanethiol (BDMT) with sulfhydryl groups to connect with Ni centers for forming S-NiBDC structure. In

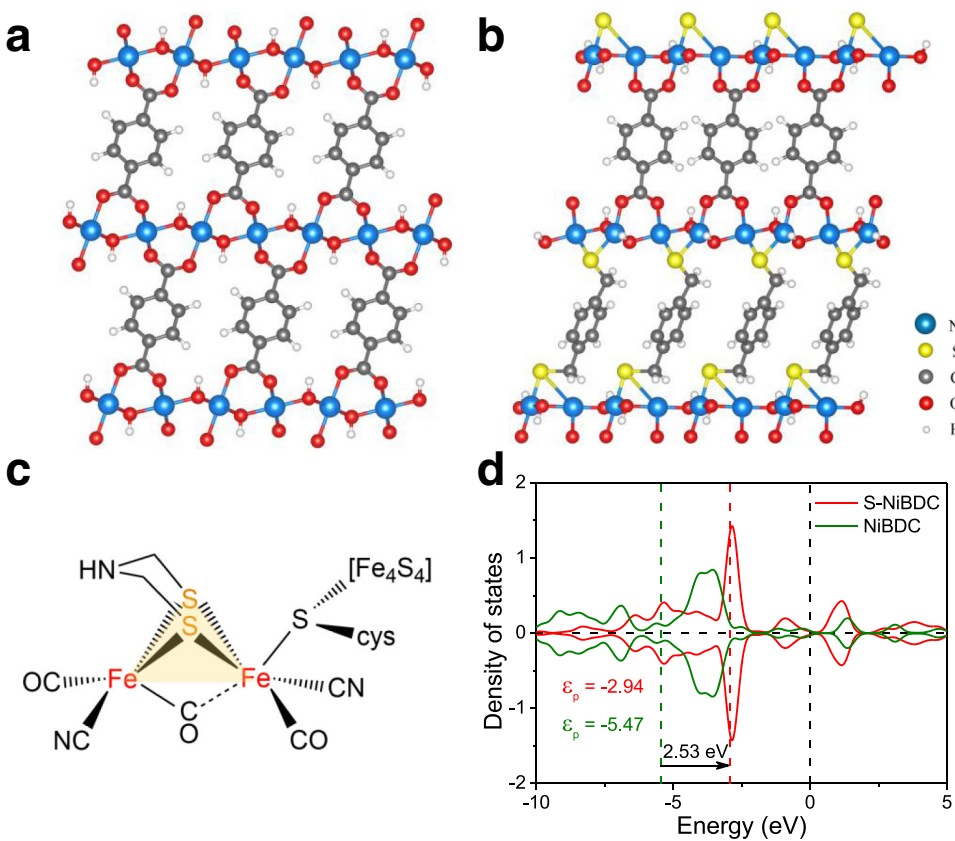

**Fig. 1 | The effect of ligand modulation for density of states by theory predictions.** Models of optimized NiBDC (**a**) and S-NiBDC (**b**) structures. **c** Structure of active site for [FeFe]-hydrogenase. **d** PDOS of p-states for NiBDC and S-NiBDC models.

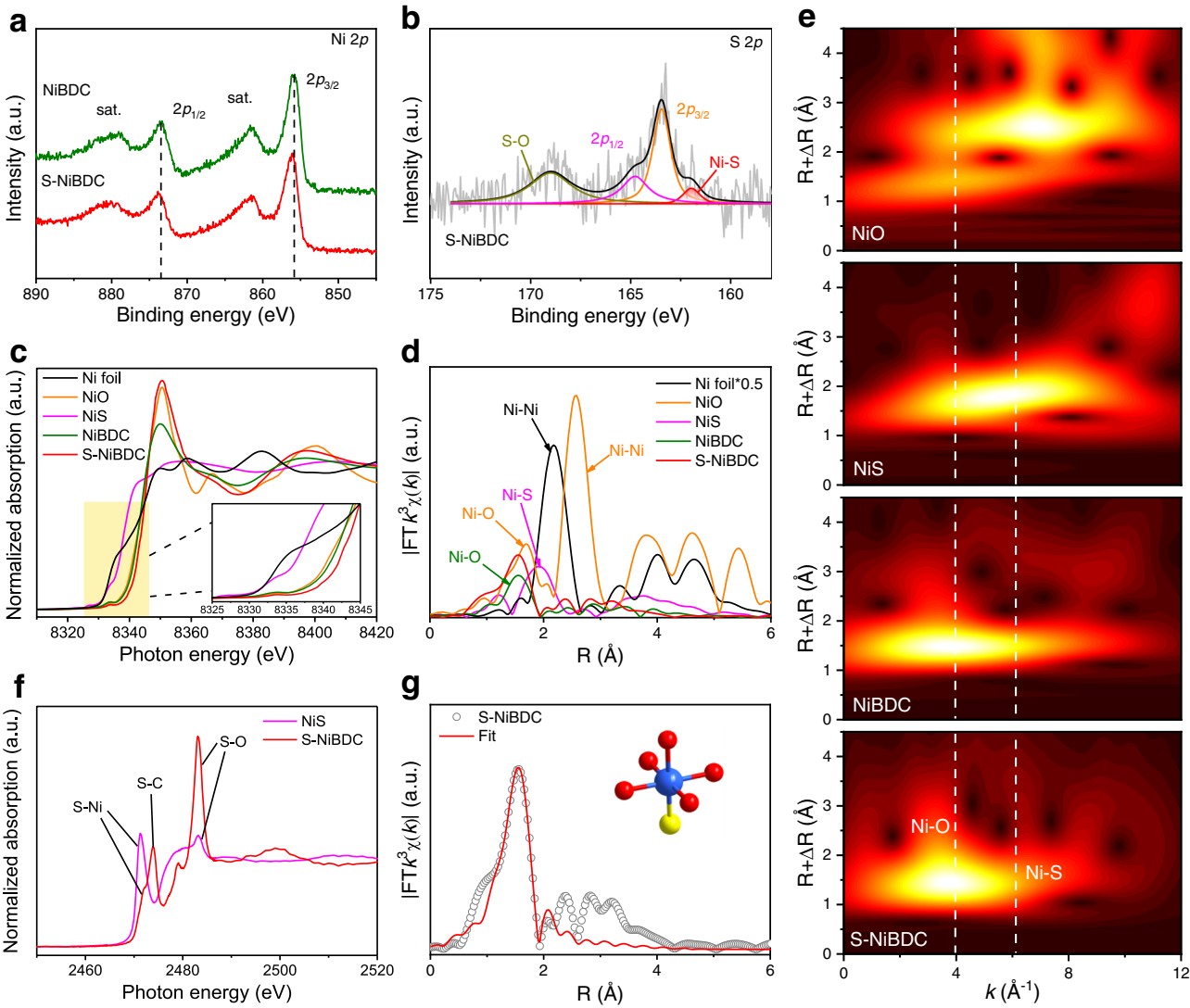

**Fig. 2 | Electronic structure characterizations. a** High-resolution XPS spectra for Ni 2*p* of S-NiBDC and NiBDC. **b** High-resolution XPS spectrum for S 2*p* of S-NiBDC. **c** Ni *K*-edge XANES and **d** $k^3$-weighted EXAFS spectra of NiBDC, S-NiBDC, Ni foil, NiO, and NiS. **e** WT-EXAFS spectra of NiBDC, S-NiBDC, NiO, and NiS. **f** S *K*-edge XANES spectra of NiS and S-NiBDC. **g** Fourier transform-EXAFS fitting results of S-NiBDC.

such S-NiBDC system, the "Ni$_2$-S$_1$" motifs are similar to the Fe-S core in [FeFe]-hydrogenase (Fig. 1c). As powerful descriptors for transition metal catalysts, the *d*- and *p*-band centers from the partial density of states (PDOS) are calculated[25]. The NiBDC (−1.16 eV) and S-NiBDC (−1.17 eV) models exhibit a similar *d*-band center ($\varepsilon_d$) (Fig. S1). After the introduction of S species, however, the *p*-band center ($\varepsilon_p$) of S-NiBDC model has been pushed closer to the Fermi level than that of NiBDC model with an increase of 2.53 eV, which indicates that the *p* states of S-NiBDC structure contribute to the high HER activity more easily (Fig. 1d)[26]. Besides, the introduced S species bring more states for the bonding and anti-bonding orbitals, which could help to the intermediates adsorption and electron transport for the whole HER reaction. These results predict that model S-NiBDC with constructed mimic hydrogenase structure possesses beneficial properties for HER.

## Materials synthesis and characterization

In light of the above theoretical results, a catalyst of S-NiBDC nanosheet arrays was synthesized by an organic linker modulated solvothermal method, as illustrated in Fig. S2a. NiCl$_2$·6H$_2$O, BDC, and BDMT were used as precursors and dissolved in a mixed solvent. Subsequently, the obtained mixed solution was sealed in an autoclave

with one piece of conductive Ni foam (NF) followed by a solvothermal reaction, in which the BDC ligands and BDMT dopants coordinate with Ni$^{2+}$ ions to form S-NiBDC. After a systematic optimization of reaction temperatures and times, the optimal synthetic condition was determined to be 150 °C and 3 h (Figs. S3 and S4).

As shown in field-emission scanning electron microscopy (FESEM) and transmission electron microscopy (TEM) images (Fig. S2b, c and Figs. S5 and S6), S-NiBDC nanosheets with an average thickness of ~16 nm were vertically grown on the surface of NF. X-ray diffraction (XRD) pattern of the S-NiBDC array revealed that the introduction of S did not break the original BDC-based MOF crystal structure with no new phase appearing (Figs. S2d and S7)[27], as also confirmed by Raman and Fourier transform infrared (FT-IR) analyses (Figs. S8 and S9). High-angle annular dark-field scanning transmission electron microscopy (HAADF-STEM) image and EDX elemental mapping showed homogeneously distributions of Ni, O, S, and C elements over the whole nanosheets (Fig. S2e–i and Fig. S10). The S content in the S-NiBDC array was measured by inductively coupled plasma-atomic emission spectrometry (ICP-AES) to be 1.67 wt.%.

To find out the impact of the S dopant on electronic and coordination structures of S-NiBDC, we performed X-ray photoelectron

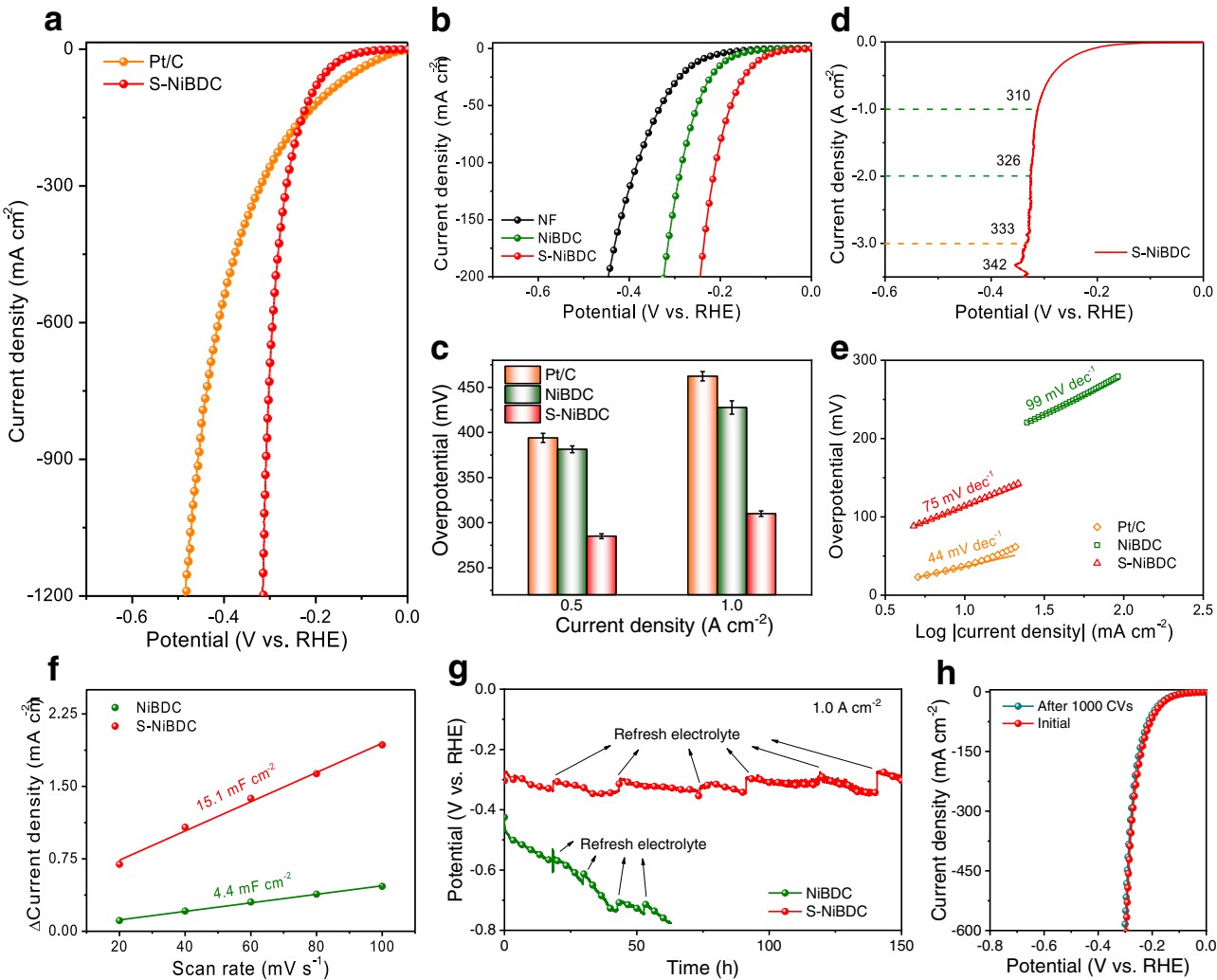

**Fig. 3 | Electrocatalytic HER performance. a** Polarization curves of S-NiBDC and Pt/C. **b** Polarization curves of NiBDC, S-NiBDC, and NF. **c** Overpotentials at 0.5 and 1.0 A cm$^{-2}$ of NiBDC, S-NiBDC, and Pt/C. Error bars correspond to the standard deviation of the three measurements. **d** Polarization curve of S-NiBDC at ampere-level current densities. **e** Tafel slopes of NiBDC, S-NiBDC, and Pt/C. **f** ECSAs of NiBDC and S-NiBDC. **g** Chronopotentiometric curves at 1.0 A cm$^{-2}$ for NiBDC and S-NiBDC. **h** Polarization curves of S-NiBDC before and after 1000 CV cycles.

spectroscopy (XPS) and X-ray absorption spectroscopy (XAS). As shown in Fig. 2a, the Ni $2p_{1/2}$ and Ni $2p_{3/2}$ peaks of S-NiBDC slightly shifted to higher binding energies by ~0.2 eV compared with that of NiBDC, suggesting the increased Ni valence state due to the change of coordination environment around center Ni atoms after introducing S species[28]. Further, in the high-resolution S $2p$ XPS spectrum, S-NiBDC displays two peaks located at 163.5 and 164.8 eV, referring to the C-S-C bonds in BDMT ligand (Fig. 2b). A small peak located at 162.0 eV assigned to Ni-S bond is observed, demonstrating that the coordination of Ni centers was modified by S-doping[29]. In the Ni $K$-edge X-ray absorption near-edge structure (XANES) spectra (Fig. 2c and Fig. S11), the oxidation state of Ni species in the S-NiBDC increases to +2.2 with respect to that of NiBDC (+1.9), indicating an increased valence state of Ni species, in consistence well with the XPS results[30]. In addition, S-NiBDC displays a higher shoulder in the white line region than that of NiBDC, implying a promoted 1 $s$ to $4p_{x,y}$ transition that could efficiently boost electrochemical activity[31]. The coordination situation for center Ni sites is illuminated by Fourier transformed extended X-ray absorption fine structure (EXAFS) spectra (Fig. 2d). For S-NiBDC, the higher peak located at 1.57 Å is attributable to the combined Ni-O and Ni-S scattering interactions in the first shell coordination (Fig. S12)[32]. Further analyses on the wavelet transform (WT) (Fig. 2e and Fig. S13) show that the maximum intensity of S-NiBDC closed to 3.9 Å$^{-1}$ is derived

from the light O atoms coordination, while the peak located at 6.1 Å$^{-1}$ can be attributed to the Ni-S bonds, indicating the predominantly O-coordinated BDC-based MOF structure with introduced S atoms as partial coordinators[33,34]. The S $K$-edge XANES spectrum of S-NiBDC displays an absorption feature centered at 2471.3 eV associated with the S-Ni species, confirming the successful coordination of Ni atoms with BDMT ligands (Fig. 2f)[35,36]. To further uncover the detailed coordination environments around the Ni center before and after S-doping, we conducted the quantitative EXAFS fitting analyses. It was found that each Ni atom in NiBDC is coordinated with six O atoms (Ni-O$_6$) in an octahedron structure (Fig. S14). After the S-doping, S atom occupies the O site to break the symmetrical Ni-O$_6$ coordination for Ni-O$_5$S$_1$ coordination, forming the local triangular "Ni$_2$-S$_1$" region in S-NiBDC by combining with two adjacent Ni atoms in Ni-oxide layers (Fig. 2g and Table S1). Since the XAS technique, a characterization for the average structural information of the detected element, is still limited in sufficiently verifying this active region in trace-doped MOF system, the direct evidence of Ni$_2$-S$_1$ region is currently lacking.

## HER performances in alkaline media

The electrocatalytic HER performance of S-NiBDC array was evaluated in a standard three-electrode system with 1.0 M KOH solution. For comparison, control NiBDC without S dopant (Fig. S15) and

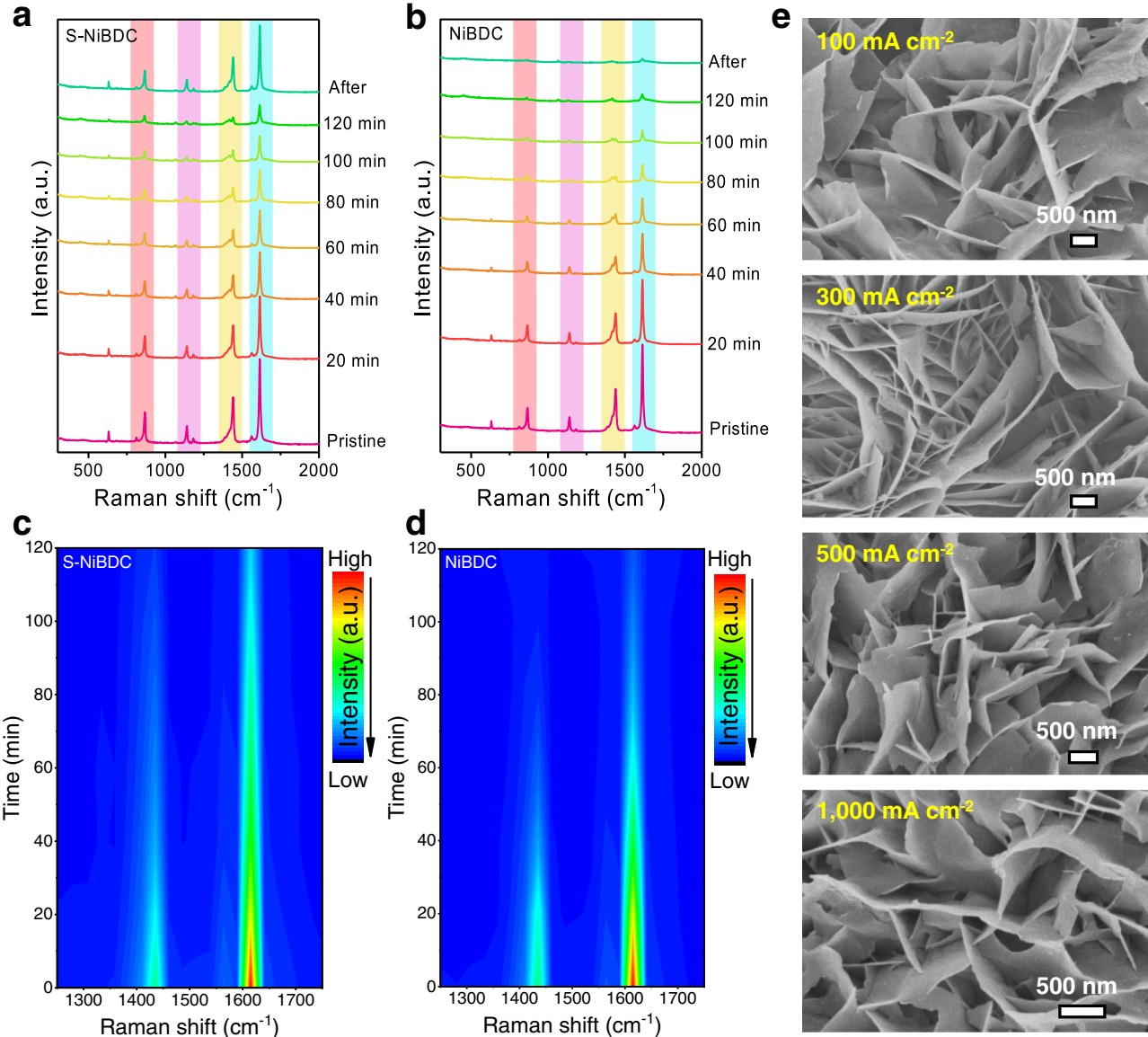

**Fig. 4 | Enhanced structural stability identified by In-situ Raman spectra and FESEM imaging. a–d** In-situ Raman spectra and corresponding contour plots of S-NiBDC and NiBDC at −0.3 V *vs.* RHE for different times. **e** FESEM images of S-NiBDC after HER tests at different current densities.

commercial Pt/C were tested under the same conditions. As can be seen in Fig. 3a, S-NiBDC can attain exceptionally high current densities, reaching 0.5 and 1.0 A cm$^{-2}$ at very low overpotentials of 285 and 310 mV, respectively, much lower than those of NiBDC (381 mV and 428 mV) and even surpass benchmark Pt/C (394 mV and 462 mV) at the same current densities (Fig. 3b, c and Fig. S16). Impressively, S-NiBDC only requires low overpotentials of 310, 328, and 342 mV to achieve higher ampere-level current densities of 1.0, 2.0, and 3.5 A cm$^{-2}$ (Fig. 3d). The enhanced HER performance for S-NiBDC is further confirmed by a smaller Tafel slope of 75 mV dec$^{-1}$, which is much lower than 99 mV dec$^{-1}$ for NiBDC (Fig. 3e) due to the favorable catalytic kinetics[37,38]. The rapid charge transfer for S-NiBDC is also supported by electrochemical impedance spectroscopy (EIS), in which the $R_{ct}$ value for S-NiBDC is smaller than that of NiBDC (Fig. S17 and Table S2). Besides, the S-NiBDC possesses a higher electrochemically active surface area (ECSA, $C_{dl}$ = 15.1 mF cm$^{-2}$) than the NiBDC (4.4 mF cm$^{-2}$), implying highly exposed active sites of S-NiBDC (Fig. 3f and Fig. S18). In addition, the comparing polarization curves normalized by ECSA for S-NiBDC and NiBDC verify the enhanced intrinsic HER activity (Fig. S19)[39]. The multi-step chronopotentiometric curve

for S-NiBDC shows that the corresponding potential is steady and responds quickly to the current density changes from 100 to 600 mA cm$^{-2}$ (Fig. S20), revealing a favorable mechanical robustness and mass transport characteristic[40]. Additionally, the negligible Faraday efficiency loss is observed for S-NiBDC during HER (Fig. S21).

**Stability evaluations in HER**
In view of the importance of catalyst stability to hydrogen production particularly at industrial-current density level, we further evaluated electrocatalytic HER stability of S-NiBDC by chronopotentiometry and long-term cycling measurements. The durability tests demonstrate that the S-NiBDC maintains its high catalytic activity over 150 h at 1.0 A cm$^{-2}$, while the HER performance of NiBDC decreases rapidly (Fig. 3g and Fig. S22). The superior robustness of S-NiBDC is also proved by 1,000 CV cycles without any obvious activity changes (Fig. 3h). Compared to most of the previously reported HER electrocatalysts (Fig. S23 and Tables S3–S5), S-NiBDC still exhibits superior alkaline HER performance with excellent stability that especially reaches ampere-level current densities (>1.0 A cm$^{-2}$), ranking the top of MOF- and sulfur-coordinating complex- based electrocatalysts.

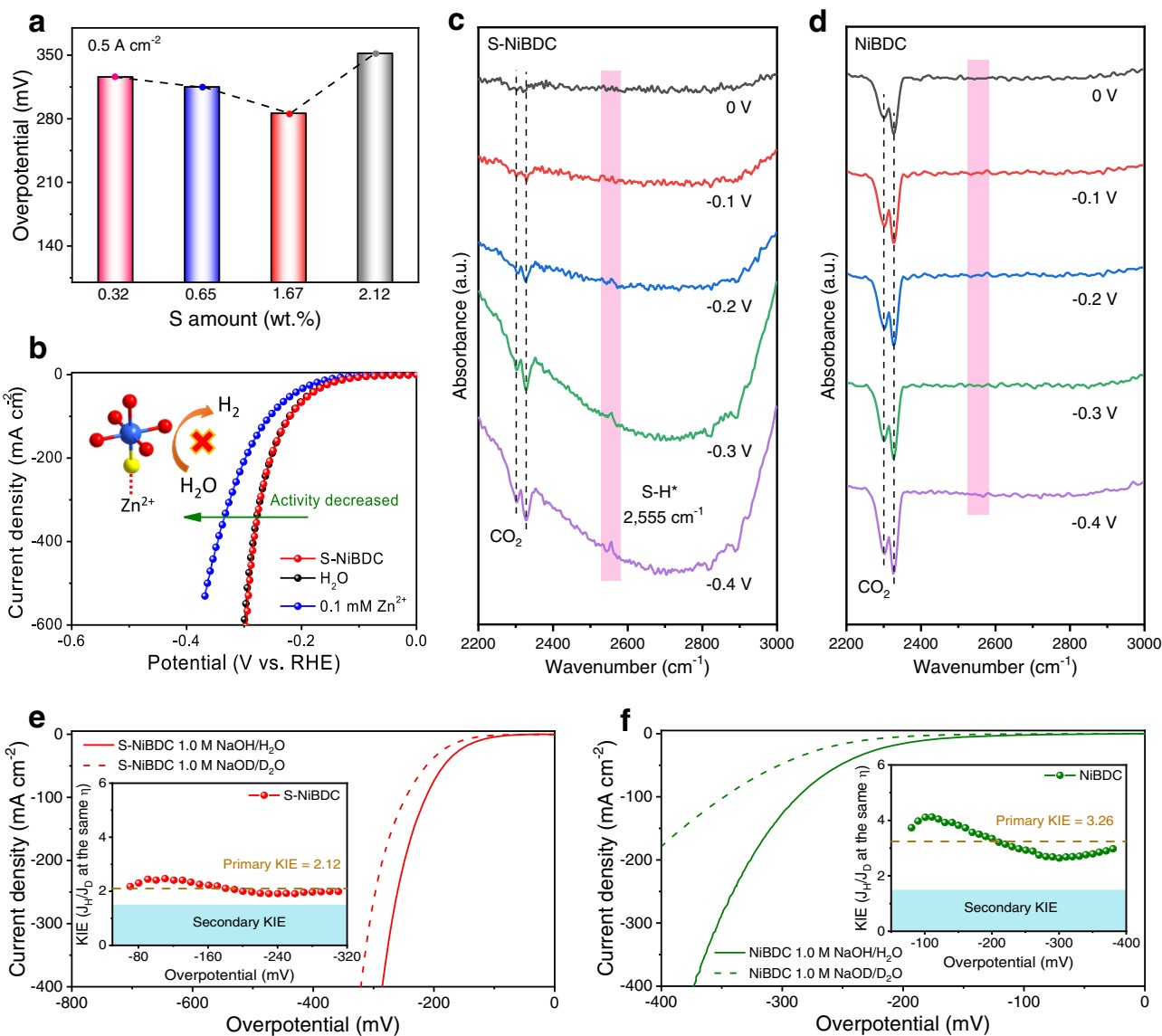

**Fig. 5 | Mechanism analyses of accelerated HER kinetics. a** Overpotentials at 0.5 A cm⁻² for S-NiBDC samples with different S doping amounts. **b** Polarization curves of S-NiBDC before and after soaking in 0.1 mM Zn(NO₃)₂ solution and H₂O. **c, d** In-situ ATR-FTIR spectra of S-NiBDC and NiBDC with the potential of 0 to −0.4 V *vs.* RHE. **e, f** Polarization curves of S-NiBDC and NiBDC in 1.0 M NaOH H₂O solution and 1.0 M NaOD D₂O solution. The insets are the KIE values *vs.* overpotential.

To better understand the enhanced catalytic stability of S-NiBDC during the high-current-density HER process, we further performed in-situ electrochemical-Raman analyses on both catalysts to probe the structure evolution under −0.3 V *vs.* RHE for different reaction times. As shown in Fig. 4a, b, both S-NiBDC and NiBDC catalysts initially show four primary peaks located at 1618, 1444, 1139, and 865 cm⁻¹, corresponding to the coordinated carboxylate groups and C-H region of the benzene rings in organic ligands within the BDC-based MOF. With the progression of reaction time, the signal intensity of the S-NiBDC decreases finitely due to the formation of H₂ bubbles, and the intensity of signal peak is rapid recovery when no external potential is applied[41]. In contrast, the signal intensity of the NiBDC drops rapidly without a significant recovery, as clearly reflected by the corresponding contour plots (Fig. 4c, d). The improved structural stability could be attributed to the higher Ni valence state in the S-NiBDC to strengthen the Ni-O bonds through the strong interaction between Ni and S atoms, thus efficiently enhancing the structural stability of BDC-based MOFs during electrocatalysis[42–44]. The enhanced structural stability of S-NiBDC is further evidenced by XRD results, which display a limited change in the

crystal structure of S-NiBDC after HER tests; in contrast, significant degradation is observed for NiBDC (Figs. S24 and S25). Further, no morphological change was observed by FESEM imaging when progressively surging the current density even up to 1.0 A cm⁻², while no dissolved organic species in the electrolyte of S-NiBDC after HER were detected[45] (Fig. 4e and Fig. S26). In contrast, NiBDC underwent severe structure collapse under same conditions, leading to a dramatic morphology deterioration (Fig. S27). Post-XPS and XAS analysis on S-NiBDC further reveal hardly changed oxidation states and coordination situations (Figs. S28–S30). Clearly therefore, the S-doping reinforces Ni-O covalency to stabilize MOF structure during HER at high current densities, thus avoiding phase transformation and morphological change.

**Origin of enhanced HER activity**

To gain more insights into the effects of the S-doping on the improved HER performance, we regulated the S-dopant content from 0.32 to 2.12 wt.% and found the optimal S content to be 1.67 wt.% (Fig. 5a and Figs. S31–S36). The poisoning experiment was further performed with

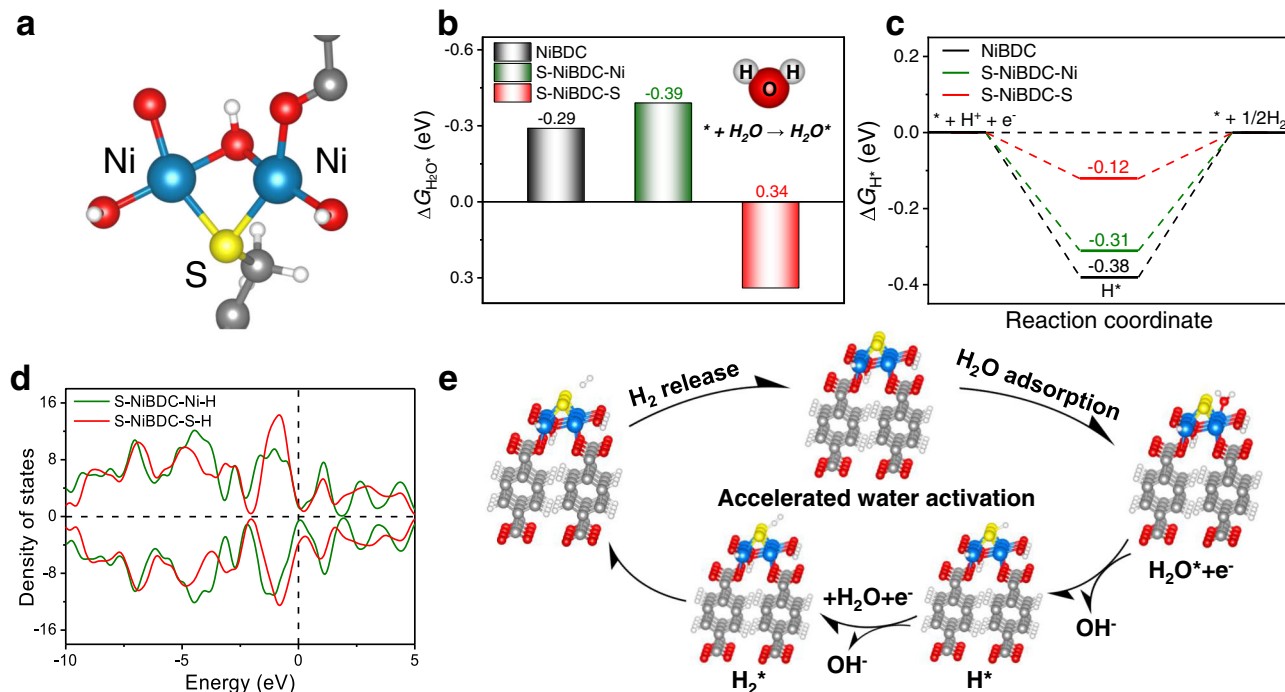

**Fig. 6 | Theoretical calculations. a** Model of "Ni$_2$-S$_1$" active region in S-NiBDC. Free-energy diagrams for **b** water adsorption and **c** H adsorption of NiBDC and S-NiBDC. **d** PDOS of different sites for S-NiBDC with H binding. **e** Schematic illustration of the HER mechanisms through Volmer-Heyrovsky pathway on S-NiBDC. Color legend for atoms: gray, C; white, H; blue, Ni; red, O; yellow, S.

Zn$^{2+}$ ions as binding species, which can block the S sites and inhibit activity[46], as evidenced by a significant decrease in the current density and overpotential shown in Fig. 5b. As expected, the HER activity of the poisoned S-NiBDC showed a negligible change after immersing in water. To further identify the definite role of S species, in-situ attenuated total reflectance infrared absorption spectroscopy (ATR-FTIR) measurements were recorded. As shown in Fig. 5c, an absorption peak located at 2555 cm$^{-1}$ was detected and gradually strengthened on S-NiBDC from 0 to −0.4 V *vs*. RHE, which was ascribed to the S-H* intermediate[47]. In contrast, there was no significant intermediate peak observed for NiBDC in the same potential window (Fig. 5d). These results demonstrate that the key S-H* intermediate was only formed on the S-NiBDC during the HER process in alkaline conditions, indicating the introduction of S active species acted as H atom catchers for facilitating the H$_2$ evolution.

Operando EIS measurements were carried out to distinguish the elementary reactions in HER. As seen in Fig. S37, in which the low and middle-frequency regions in the bode plots are related to the Volmer and Heyrovsky steps, both S-NiBDC and NiBDC followed a Volmer-Heyrovsky mechanism. Compared with NiBDC, S-NiBDC showed a faster decrease in the low-frequency region with increasing over-potential, demonstrating the Volmer step as the rate-determining step (RDS) during alkaline HER that was accelerated on S-NiBDC. In addition, we investigated the deuterium kinetic isotope effects (KIE) of the HER (Fig. 5e, f). The KIE studies reflect the proton transfer kinetics involved in the water dissociation process with the cleavage of HO-H bond, which is the main barrier for the restrained Volmer step. As for S-NiBDC, the primary KIE value in HER overpotential regions fluctuated around 2.12, much lower than that of NiBDC (primary KIE = 3.26), indicating the water dissociation kinetics for S-NiBDC was apparently accelerated, in good consistence with Tafel analyses[48–50]. According to the above results, we can therefore conclude that the introduction of S species not only help to accelerate water dissociation kinetics, but also adsorb with H atom through S-H* bond to facilitate the kinetics of alkaline HER process.

## Theoretical insights of the underlying mechanism

The above experimental results are further complemented by DFT calculations of the S-NiBDC and NiBDC models to gain a better understanding of the HER mechanism. Figure 6a shows the locally modified triple-atom "Ni$_2$-S$_1$" active region for S-NiBDC, which is composed of two Ni atoms and one S atom. It was found that the Ni sites modified by the S atom are thermodynamically more favorable for water adsorption among all the investigated sites, indicating that water molecule is easily activated on the Ni sites in S-NiBDC for the accelerated HER process (Fig. 6b)[51]. Figure 6c shows the calculated adsorption Gibbs free energies of H* ($\Delta G_{H^*}$), the value of which should be close to zero for ideal HER catalyst[52]. Compared with the NiBDC (−0.38 eV) as well as the Ni site (-0.31 eV) in S-NiBDC, the S site in S-NiBDC delivers the optimal value of $\Delta G_{H^*}$ (−0.12 eV) that is closer to the optimal value, revealing that the S site in the active region is most effective for H$_2$ evolution. As displayed in Fig. 6d, the introduced S sites of S-NiBDC binding with H atom show that the total charge transport capacity of the system is enhanced, and the upper and lower spins are closer to the Fermi level. The lower spin demonstrates more states on the Fermi level, which indicates the S site in S-NiBDC model has more hybridization with the H atom (Figs. S38 and S39) than the Ni site. Meanwhile, this hybridization brings the better electron transport performance for the catalyst and reaction process which could be one of the reasons to boost the HER. DFT calculations are consistent well with the in-situ ATR-FTIR measurements, manifesting that the intro-duced S site can integrate with H atom to generate S-H* intermediate for the subsequent Heyrovsky step. Based on these experimental and theoretical results, we can describe the whole HER process in the tri-angular "Ni$_2$-S$_1$" active region as follows: water molecules are trapped by the S-coordinated Ni sites, followed by dissociation on the Ni sites to form H atom and OH$^-$ for activating the water molecules with the aid of one free electron (Fig. 6e and Fig. S40); the generated H atom is further adsorbed on the S site and eventually produce H$_2$ (Fig. S41).

Motivated by such a high HER performance of S-NiBDC, an alkaline Zn-H$_2$O fuel cell is assembled to integrate H$_2$ and electricity production

simultaneously (Fig. S42)[53]. A larger current density is found from the discharging polarization curve when S-NiBDC acts as the cathode compared with NiBDC counterpart. S-NiBDC can reach a maximum power density of 4.5 mW cm$^{-2}$, much higher than that of NiBDC (3.1 mW cm$^{-2}$). The Zn-H$_2$O cell responds rapidly and is stable over the current densities from 10 to 50 mA cm$^{-2}$. At 10 mA cm$^{-2}$, this Zn-H$_2$O cell shows good durability over 10 h. To demonstrate practical potential, we integrate three Zn-H$_2$O cells in series to light a red light-emitting diode.

## Discussion

In summary, we have successfully developed highly active S-NiBDC nanosheet arrays with constructed "Ni$_2$-S$_1$" active centers via a linker modulation strategy for efficient alkaline HER. Benefiting from the $p$-band center regulation and enhanced electron transfer, the S-NiBDC showed a superb and robust HER activity in base with a low over-potential of 310 mV at 1.0 A cm$^{-2}$ and a small Tafel slope of 75 mV dec$^{-1}$, outperformed most of the reported MOF-based electrocatalysts and even surpassed the state-of-the-art Pt/C. The extraordinary electro-catalytic performance and structural stability of S-NiBDC are attributed to the formed local "Ni$_2$-S$_1$" triple-sites with exceptional electrocatalytic activity and structural stability. This S-ligand modulation strategy is highly general, which could be further extended to develop other S-MBDC catalysts (e.g., S-CoBDC and S-FeBDC) with triangular M$_2$-S$_1$ active regions. Our preliminary results confirmed that the alkaline HER activities of CoBDC or FeBDC can indeed be improved by introducing S species to form "Co$_2$-S$_1$" and "Fe$_2$-S$_1$" active regions (Figs. S43–S49). Experimental and theoretical combined studies revealed that the S-modified Ni sites accelerate water activation kinetics while the S site in the triangular "Ni$_2$-S$_1$" structure acts as the active center for sub-sequent H$_2$ production, ultimately boosting the overall HER perfor-mance. This work provides a promising universal approach for rationally designing highly active and durable MOF-based catalysts with triangular active regions for various electrochemical applications.

## Methods

### Synthesis of S-NiBDC
119 mg of NiCl$_2$·6H$_2$O, 83 mg of BDC, and 4.3 mg of BDMT were dissolved in a mixed solvent of water, ethanol, and N, N-dimethylformamide (DMF) (volume ratio = 1:1:16 mL), and stirred for 20 min to form a homogeneous solution. The obtained solution was transferred into a Teflon-lined autoclave with a piece of NF (1 cm × 3 cm) and then heated at 150 °C for 3 h in an oven, followed by naturally cooled down to room temperature. After being repeatedly washed with DMF, ethanol, and deionized (DI) water and dried in an oven at 60 °C, a final product of S-NiBDC grown on NF was obtained. The loading amount of S-NiBDC on the NF was determined to be about 1.1 mg cm$^{-2}$.

### Synthesis of NiBDC
119 mg of NiCl$_2$·6H$_2$O and 83 mg of BDC were dissolved in a mixed solvent of water, ethanol, and DMF (volume ratio = 1:1:16 mL), and stirred for 20 min to form a homogeneous solution. The obtained solution was transferred into a Teflon-lined autoclave with a piece of NF (1 cm × 3 cm) and then heated at 150 °C for 3 h in an oven, followed by naturally cooled down to room temperature. After being repeatedly washed with DMF, ethanol, and DI water and dried in an oven at 60 °C, a final product of NiBDC grown on NF was obtained. The loading amount of NiBDC on the NF was determined to be about 1.3 mg cm$^{-2}$.

### Characterization
The morphologies of the as-prepared catalysts were examined by FESEM (Hitachi SU-8010) with EDX (Oxford, X-max80), TEM (HT7700), and HRTEM (Tecnai G2 F20 S-TWIN). The crystal structures of the as-prepared catalysts were measured by XRD (ZETIUM DY powder XRD unit) using Cu Kα radiation at 4.0 KW. The chemical environments of

the as-prepared catalysts were measured by XPS (Escalab 250Xi) with Al Kα radiation. All XPS spectra were calculated after correction with C 1 s peak at 284.8 eV. Raman spectra of the as-prepared catalysts were measured with a HORIBA/XploRA PLUS at 532 nm. FT-IR spectra were recorded on ThermoFisher Nicolet 6700 at room temperature. The contents of nickel and sulfur in the as-prepared catalysts were analyzed by ICP-AES (Agilent 720).

### Electrochemical measurements
An electrochemical analyzer (CHI 760E) was used to perform all elec-trochemical experiments in a standard three-electrode configuration. A saturated Ag/AgCl electrode and a graphite rod were used as a reference and counter electrode, respectively. The potential was converted to reversible hydrogen electrode (RHE) via a Nernst equa-tion ($E_{RHE} = E_{Ag/Ag/Cl} + 0.059 \times pH + 0.197$). The scan rate of linear sweep voltammetry (LSV) was set to 5.0 mV s$^{-1}$ with potentials between 0.233 and −0.577 V $vs.$ RHE in 1.0 M KOH electrolyte to investigate the HER activities of the as-prepared catalysts. EIS test was measured at −0.177 V $vs.$ RHE with a frequency range from 10$^5$ to 0.01 Hz. All polarization curves were calibrated with the iR correction. CV cycles (CVs) with the scan rates from 20 to 100 mV s$^{-1}$ were applied to analyze ECSA.

### In-situ electrochemical-Raman tests
The in-situ electrochemical-Raman were carried out in a custom-made spectro-electrochemical cell, using as-prepared catalysts as the work-ing electrode, Pt wire as the counter electrode, and saturated Ag/AgCl electrode as the reference electrode. Each Raman spectrum was col-lected at 532 nm.

### In-situ ATR-FTIR experiments
In-situ ATR-FTIR were carried out on an Invenio-R spectrometer equipped with a mercury cadmium telluride (MCT) detector cooled with liquid nitrogen The Au-coated Si hemispherical prism was used as the conductive substrate for catalysts and the IR refection element. Pt electrode and Ag/AgCl electrode were used as counter electrode and reference electrode, respectively. All spectra were presented in absorbance, Abs = −log(R/R$_0$) (R: reflectance of the sample spectrum, R$_0$: background spectrum).

### Calculation method
To apply first-principles-based quantum mechanics simulations, all density functional theory (DFT) computations were done via the plane-wave pseudopotential approach. The spin-polarized generalized gra-dient approximation (SGGA) was built up as an exchange-correlation function within the Perdew-Burke-Ernzerhof (PBE) form in this study. A 3 × 3 × 3 Monkhorst-Pack $k$-points grid was used for $k$-points sampling, and 900 eV plane-wave expansion was setup for energy cut-offs. The Ni-3$d^8$4$s^2$, O-2$s^2$2$p^4$, C-2$s^2$2$p^2$, S-3$s^2$3$p^4$ and H-1$s^1$ were setup for valence states. A vacuum slab of about 15 Å was maintained in the super-cell configuration that was large enough for the calculations. The SCF tolerance for geometrical optimization and phonon calculations was <1e$^{-6}$ eV·atom$^{-1}$. The more computational details can be found in the Supporting Information.

## Data availability
The data that support the findings of this study are available from the Source Data. Additional data are available from the corresponding authors upon reasonable request. Source data are provided with this paper.

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

## Acknowledgements

Y.H. acknowledges the financial support from National Natural Science Foundation of China (22278364, 22211530045, 21922811, 21878270, 22178308), the Zhejiang Provincial Natural Science Foundation of China (LR19B060002), the Fundamental Research Funds for the Central Universities (226-2022-00055), Zhejiang University Global Partnership Fund, the Startup Foundation for Hundred-Talent Program of Zhejiang University. M.Q. acknowledges the financial support from National Natural Science Foundation of China (U20A20246), the Fundamental Research Funds for the Central Universities. The XAS experiments were performed at the BL14W1 station in Shanghai Synchrotron Radiation Facility, the 1W1B station in Beijing Synchrotron Radiation Facility, and BL17C station in Taiwan Synchrotron Radiation Research Center.

## Author contributions

F.C. and Y.H. conceived the idea, and F.C. performed experiments. C.-L.D., J.-L.C., and L.-C.H. performed XAS characterizations. L.H. and M.Q. performed DFT calculations. Q.Z. performed HAADF-STEM tests. X.P., B.Y., Z.L., and L.L. provided helpful suggestions. F.C., Y.H., and L.D. wrote the manuscript. All authors discussed the results and commented on the manuscript.

## Competing interests

The authors declare no competing interests.
