## [Peer review file · Nature Communications]

Reviewers' comments:

Reviewer #1 (Remarks to the Author):

This work by Cheng et al. developed a ligand regulation method to synthesized S-doped Ni-benzenedicarboxylic acid (BDC)-based MOF nanosheet as electrocatalysts for hydrogen evolution reaction (HER). The HER activity and stability of the S-doped catalysts (S-NiBDC) were significantly improved compared to the catalysts without S doping (NiBDC), the overpotential to obtain 1 A cm⁻² of S-NiBDC catalyst is even lower than that on benchmark-catalyst Pt/C. Combining in situ Raman investigations, operando EIS measurements and DFT calculations, the authors concluded that the S species can stabilize the catalyst structure and facilitate water dissociation kinetics. Detailed characterizations and careful analysis were made in this manuscript, and the performance is excellent, however, some conclusions cannot be well supported by the Raman investigations. In addition, the DFT calculation is not well performed to support the arguments. This work needs substantial improvement before it can meet the standard of Nature Communications.

1. The observed S-H* intermediate only exists at potentials lower than -0.8 VRHE in Raman experiments, while the potentials used for activity measurements are much higher (> -0.4 VRHE), thus it is inappropriate to correlate the HER activity with the S-H* intermediate due to the inconsistency of the potential windows. In addition, the assignment of S-H* intermediate is lack of any evidence. Corresponding references or DFT calculations should be given to support the assignment.

2. Similar to question #1, it is also inappropriate to investigate the stability of the catalyst structure by applying such negative potentials (up to -1.2 VRHE) in Raman experiments. Tracking the peak intensity with time at a certain potential used for reactivity test to study the stability of catalyst structure is more convincing.

3. The DFT results in Figure 6 are quite confusing and the authors are suggested to double check the DFT data processing and provide more computational details, such as what is the electrode potential applied in the calculations.

(1) The free energies of initial state and final state in Figure 6(b) should not be the same.

(2) Is the initial state in Figure 6(c) correct? It should be H₂O*.

(3) The authors should be more careful in distinguishing between the terms of “free energy changes” and “energy barrier”. In Figure 6(c), 2.51 eV should be the energy barrier of water activation instead of “free energy” that the authors used in line 229. In addition, the “potential barrier” in line 236 is exact the “free energy changes” according to Figure 6(d).

(4) If the energy barrier of water activation on S-NiBDC is as high as 2.51 eV, the statement that “S-NiBDC delivers the lowest potential barrier (1.19 eV)” in line 236 is incorrect.

4. The current density (100 mA cm⁻²) used for stability study is too low compared to the achieved highest current density (3.5 A cm⁻²), and is also much lower than that in previous studies (> 400 mA cm⁻²) (Adv. Energy Mater. 2018, 8, 1801065; Appl. Catal. B 2019, 258, 118023; Nano Energy 2019, 57, 1-13). A higher current density for the long-term electrolysis is required in order to better illustrate the stability of the catalyst.

Reviewer #2 (Remarks to the Author):

The authors present a mixed NiBDC/BDMT MOF for the reduction of protons in an alkaline solution. The use of -S- moieties to enhance conductivity is not a new phenomenon (see the work of Dinca and Marinescu) and more importantly, Marinescu used 2D NiTHT (and other derivatives) for the reduction of protons. The Marinescu materials achieve higher current densities at lower overpotential than those reported in the submitted manuscript (although notably under acidic conditions). The authors do provide a comparison to other MOF approaches - presumably tested under alkaline conditions - and as shown, it does not have the leading performance.

The authors indicate in the opening paragraph that Pt is used in alkaline HER - this is certainly not true. Industrial electrolyzers use NiFe electrodes for proton reduction. The reviewer suggests the authors compare the modified MOF to NiFe to increase the relevance of controls.

The authors should normalize their HER rates to ESCA. Are there still differences between the two configurations - BDC only vs. mixed BDC/BDMT. The reviewer imagines so, however, it would be good to see in the SI.

The stability of the mixed NiBDC/BDMT material is questionable. It is certainly more stable than BDC, but there is a clear loss of crystallinity in the PXRD for the mixed MOF. Beyond PXRD and SEM, was any additional post-catalysis characterization completed? Are there dissolved species in the electrolysis solution (NMR or ICP-MS)? Is the distribution of Ni species unchanged (XPS)?

The reviewer appreciates the deep mechanistic discussion, including the characterization of the S-H* intermediate. This result is interesting, although potentially expected given previous studies with dithiolene compounds, where the protonation of the S or N linkage is preferred to metal binding. That

said, how does the S-H bond formation impact the local geometry within the MOF. This comment refers back to the stability paragraph - in previous non-innocent ligand arrangements protonation at a S or N resulted in bond breakage between the node and linker, and thus, a pathway for instability.

The characterization of the MOFs themselves seems relatively standard. The PXRD patterns do seem to exhibit a high background. Could a zero-background holder be used to eliminate this noise?

Reviewer #3 (Remarks to the Author):

The authors of this paper report the fabrication of S-doping Ni-benzenedicarboxyl acid (BDC)-based metal-organic framework (S-NiBDC) as a hydrogen evolution (HER) electrocatalyst in alkaline solution. By theoretical calculations and various structural characterizations, it is claimed that the triangular “Ni₂-S₁” motif in S-NiBDC catalysts could facilitate the water activation kinetics for boosting high HER activity. However, the concept of dopant-regulated BDC-based catalysts toward improved HER performance has been reported before. Moreover, the synthetic approach of S-NiBDC catalysts in this work seems very routine and their structures are questionable. Based on these considerations, this reviewer fails to recommend the publication of this manuscript in Nature Communications. I hope the following comments will be helpful for improving the quality of this manuscript.

1. The proposed structural model of S-NiBDC in Figure 1b is not consistent well with the practical catalysts. There should be the coexistence of benzenedicarboxyl and 1,4-benzenedimethanethiol ligands in the S-NiBDC catalysts, rather than the presence of only 1,4-benzenedimethanethiol linkers. Moreover, in the S-NiBDC model, partial S atoms bind with two Ni atoms and the others bind with only one Ni atom. Why?
2. Based on the p band center of S-NiBDC shown in Figure 1d, it is claimed that S-NiBDC possesses beneficial properties for HER. But the p band center of S-NiBDC is located at -2.94 eV, which is still very far from the Fermi level.
3. The average metal oxidation states of Ni species in both NiBDC and S-NiBDC catalysts are recommended to be given, based on the results of XANES and XPS spectra.
4. The XAFS fitting results shown in Table S1 can not clearly support the formation of triangular “Ni₂-S₁” region in the S-NiBDC catalysts. Is there any evidence to confirm this point?

5. Moreover, the WT-EXAFS plots of NiBDC and S-NiBDC catalysts look similar to each other (Figure 2e). Since the Ni-S bonds are claimed to form in S-NiBDC catalyst, there should be some difference between the WT-EXAFS plots of NiBDC and S-NiBDC samples.

6. In general, the pure NiBDC catalyst exhibits poor electrochemical HER performance. However, in this work, the HER activity of NiBDC sample is superior to that of the benchmark Pt/C catalysts. What is the reason for this?

7. To solidly demonstrate the structural stability, XAFS spectroscopy is suggested to be performed for the S-NiBDC catalysts after a long-term HER operation in alkaline.

8. As the in-situ Raman spectra shown in Figure 5c, the vibrational signal of S-H* bonds can be obtained till the potentials of -1.0 V or -1.2 V is applied. It is claimed that the S-NiBDC catalyst deliver high current density of 1 A cm⁻² at a relatively low overpotential of 310 mV. Why the vibrational signal of S-H* bond is undetectable at a low potential, such as -0.4 or -0.5 V?

9. Based on the Gibbs free-energy calculations in Figure 6d, the formation of *H on the S site of S-NiBDC model has the lowest potential barrier of 1.19 eV, which is still much larger than the optimal value for HER. The Gibbs free energy change of Pt catalysts during HER is suggested to be provided here as reference.

Response to Reviewers' Comments

Itemized Response to the Comments on Nature Communications (Manuscript number: NCOMMS-22-14896).

For your reference, the detailed point-by-point response is listed below. To save your and the reviewers' time to quickly read what we have done for the revision, in this response letter, the texts for “**Authors' Response**” are marked in blue font, and the changes are highlighted in **YELLOW** color in both revised manuscript and Supporting Information.

Response to the comments of Reviewer #1

Comments: *This work by Cheng et al. developed a ligand regulation method to synthesized S-doped Ni-benzenedicarboxylic acid (BDC)-based MOF nanosheet as electrocatalysts for hydrogen evolution reaction (HER). The HER activity and stability of the S-doped catalysts (S-NiBDC) were significantly improved compared to the catalysts without S doping (NiBDC), the overpotential to obtain 1 A cm^{-2} of S-NiBDC catalyst is even lower than that on benchmark-catalyst Pt/C. Combining in situ Raman investigations, operando EIS measurements and DFT calculations, the authors concluded that the S species can stabilize the catalyst structure and facilitate water dissociation kinetics. Detailed characterizations and careful analysis were made in this manuscript, and the performance is excellent, however, some conclusions cannot be well supported by the Raman investigations. In addition, the DFT calculation is not well performed to support the arguments. This work needs substantial improvement before it can meet the standard of Nature Communications.*

Authors' Response:

Thank you very much for all of the comments on our manuscript. As shown in the following responses, we have supplemented more detailed and systematic data in the revised manuscript to make our work more solid. We hope that the reviewer finds our manuscript suitable for the publication in Nature Communications.

Comment 1: *The observed S-H* intermediate only exists at potentials lower than -0.8 V RHE in Raman experiments, while the potentials used for activity measurements are much higher ($> -0.4 \text{ V RHE}$), thus it is inappropriate to correlate the HER activity with the S-H* intermediate due to the inconsistency of the potential windows. In addition, the assignment of S-H* intermediate is lack of any evidence. Corresponding references or DFT calculations should be given to support the assignment.*

Authors' Response:

We appreciate the valuable comment from the reviewer. In the previous version, the inconsistent Raman signal may be attributed to the fact that surface-enhanced Raman spectroscopy that requires Au or Ag as a

substrate to enhance the Raman signal from chemical species close to the surface was not used (*Angew. Chem. Int. Ed.* 2021, 60, 19774). As a result, the peak of S-H* intermediate by increasing the applied potential window and the corresponding tested HER potential range are not a good match. Following the reviewer's kind suggestion, we supplemented the *in-situ* attenuated total reflectance infrared absorption spectroscopy (ATR-FTIR) for S-NiBDC and NiBDC samples with the ultra-thin Au foil deposited on the silicon to enhance the signal of S-H* intermediate. As shown in **Figure 5c**, an absorption peak located at 2555 cm⁻¹ was observed for S-NiBDC at -0.2 V vs. RHE and gradually strengthens in intensity with the applied potential increased to -0.4 V vs. RHE, which can be attributed to the formed S-H* intermediates during the HER process (*Angew. Chem. Int. Ed.* 2021, 60, 7251). However, as for NiBDC, no S-H* vibration absorption can be found at the same applied potential window (**Figure 5d**). These consequences reveal that the S species in S-NiBDC is the catalytic active sites and acts as the H atom catcher for further Heyvosky step to produce H₂.

Figure 5. (c-d) *In-situ* ATR-FTIR spectra of S-NiBDC and NiBDC with the potential of 0 to -0.4 V vs. RHE.

Corresponding revisions:

The results and discussion part

In response to this comment, we updated **Figure 5c-d** and revised the corresponding part on **Page 10** of the revised manuscript as follows:

To further identify the definite role of S species, *in-situ* attenuated total reflectance infrared absorption spectroscopy (ATR-FTIR) measurements were recorded. As shown in Figure 5c, an absorption peak located at 2555 cm⁻¹ was detected and gradually strengthened on S-NiBDC from 0 to -0.4 V vs. RHE, which was ascribed to the S-H* intermediate⁴⁷. In contrast, the vibration absorption band of S-H* intermediate was not found for NiBDC in the same potential window (Figure 5d). These results demonstrate that the key S-H* intermediate

was only formed on the S-NiBDC during the HER process in alkaline conditions, indicating the introduction of S active species acted as H atom catchers for facilitating the H₂ evolution.

Comment 2: *Similar to question #1, it is also inappropriate to investigate the stability of the catalyst structure by applying such negative potentials (up to -1.2 V RHE) in Raman experiments. Tracking the peak intensity with time at a certain potential used for reactivity test to study the stability of catalyst structure is more convincing.*

Authors' Response:

We appreciate the valuable comment. We agree well with your comments that tracking the peak intensity with time at a certain potential is more convincing to explore the structure evolution of catalysts. Following the reviewer's suggestion, we remeasured the *in-situ* electrochemical-Raman spectra at a certain potential of -0.3 V vs. RHE to investigate the structural stability of S-NiBDC and NiBDC. As shown in **Figure 4a-d**, as reaction time increases to 120 min, there is a limited decrease in Raman peak intensity of S-NiBDC due to the formation of hydrogen bubbles, however the intensity of signal peak is rapid recovery without applying potential. In comparison, the signal intensity of the NiBDC drops rapidly without a significant recovery. These results demonstrate the certainly improved stability of BDC-based MOF structure after S introduction.

Figure 4. (a-d) *In-situ* Raman spectra and corresponding contour plots of S-NiBDC and NiBDC at -0.3 V vs. RHE for different times.

Corresponding revisions:

The results and discussion part

In response to this comment, we updated **Figure 4a-d** and revised the corresponding part on **Page 9** of the revised manuscript as follows:

we further performed *in-situ* electrochemical-Raman analyses on both catalysts to probe the structure evolution under -0.3 V vs. RHE for different reaction times. As shown in Figure 4a-b, both S-NiBDC and NiBDC catalysts initially show four primary peaks located at 1618 , 1444 , 1139 , and 865 cm^{-1} , corresponding to the coordinated carboxylate groups and C-H region of the benzene rings in organic ligands within the BDC-based MOF. With the progression of reaction time, the signal intensity of the S-NiBDC decreases finitely due to the formation of H_2 bubbles, and the intensity of signal peak is rapid recovery when no external potential is applied⁴¹. In contrast, the signal intensity of the NiBDC drops rapidly without a significant recovery, as clearly reflected by the corresponding contour plots (Figure 4c-d).

Comment 3: The DFT results in Figure 6 are quite confusing and the authors are suggested to double check the DFT data processing and provide more computational details, such as what is the electrode potential applied in the calculations.

(1) The free energies of initial state and final state in Figure 6(b) should not be the same.

(2) Is the initial state in Figure 6(c) correct? It should be H_2O^* .

(3) The authors should be more careful in distinguishing between the terms of “free energy changes” and “energy barrier”. In Figure 6(c), 2.51 eV should be the energy barrier of water activation instead of “free energy” that the authors used in line 229. In addition, the “potential barrier” in line 236 is exact the “free energy changes” according to Figure 6(d).

(4) If the energy barrier of water activation on S-NiBDC is as high as 2.51 eV, the statement that “S-NiBDC delivers the lowest potential barrier (1.19 eV)” in line 236 is incorrect.

Authors' Response:

We appreciate the valuable comment from the reviewer. Following the reviewer's suggestion, we have revised the relative presentations and rearranged the **Figure 6** in the revised manuscript and supporting information. Further, the additional free energy of water adsorption was calculated, and the obtained results showed that the Ni sites with S-coordinated in S-NiBDC had the strongest ability for water adsorption, which benefits for the water activation to generate more H atoms (**Figure 6b**) (*Adv. Mater.* 2021, 33, 2106781; *Angew. Chem. Int. Ed.* 2019, 58, 4679; *Nat Commun* 2021, 12, 1369). The calculated free-energy diagrams of Volmer-Heyrovsky pathway for HER indicated that the S site in the active region is most effective for H^* intermediate to produce H_2 (**Figure 6c**).

Figure 6. (a) Model of “Ni₂-S₁” active region in S-NiBDC. Free-energy diagrams for (b) water adsorption and (c) Volmer-Heyrovsky mechanisms of different reaction sites in NiBDC and S-NiBDC ($U = 0.00$ V). (d) PDOS

of different sites for S-NiBDC with H binding. (e) Schematic illustration of the accelerated water activation for HER in S-NiBDC (right) and NiBDC (left). Color legend for atoms: gray, C; blue, Ni; red, O; yellow, S.

Corresponding revisions:

The results and discussion part

In response to this comment, we updated **Figure 6** and revised the corresponding part on **Page 11-12** of the revised manuscript as follows:

It was found that the Ni sites modified by the S atom are thermodynamically more favorable for water adsorption among all the investigated sites, indicating that water molecule is easily activated on the Ni sites in S-NiBDC for the accelerated HER process (Figure 6b)⁵¹. Figure 6c shows the Volmer-Heyrovsky pathway for HER. Compared with the Ni site (1.20 eV) in NiBDC as well as the Ni site (0.83 eV) in S-NiBDC, the S site delivers the lowest free energy changes (0.35 eV)⁵², revealing that the S site in the active region is most effective for H* intermediate.

Comment 4: *The current density (100 mA cm⁻²) used for stability study is too low compared to the achieved highest current density (3.5 A cm⁻²), and is also much lower than that in previous studies (> 400 mA cm⁻²) (Adv. Energy Mater. 2018, 8, 1801065; Appl. Catal. B 2019, 258, 118023; Nano Energy 2019, 57, 1-13). A higher current density for the long-term electrolysis is required in order to better illustrate the stability of the catalyst.*

Authors' Response:

We appreciate the valuable comment from the reviewer. Following the reviewer's suggestion, additional stability measurements under a higher current density of 1.0 A cm⁻² for S-NiBDC and NiBDC were conducted to evaluate the stability. As shown in **Figure 3f**, S-NiBDC exhibits superior stability over 150 h at 1.0 A cm⁻².

Figure 3. (f) Chronopotentiometric curves at 1.0 A cm⁻² for NiBDC and S-NiBDC.

Corresponding revisions:**The results and discussion part**

In response to this comment, we updated **Figure 3f** and revised the corresponding part on **Page 9** of the revised manuscript as follows:

The durability tests demonstrate that the S-NiBDC maintains its high catalytic activity over 150 h at 1.0 A cm⁻², while the HER performance of NiBDC decreases rapidly (Figure 3f and Figure S22).

Thanks again for all your valuable comments to help us improve the quality of our manuscript!

Response to the comments of Reviewer #2

Comment 1: *The authors present a mixed NiBDC/BDMT MOF for the reduction of protons in an alkaline solution. The use of -S- moieties to enhance conductivity is not a new phenomenon (see the work of Dinca and Marinescu) and more importantly, Marinescu used 2D NiTHT (and other derivatives) for the reduction of protons. The Marinescu materials achieve higher current densities at lower overpotential than those reported in the submitted manuscript (although notably under acidic conditions). The authors do provide a comparison to other MOF approaches - presumably tested under alkaline conditions - and as shown, it does not have the leading performance.*

Authors' Response:

We appreciate the valuable comment from the reviewer.

In the above-mentioned works of Dinca and Marinescu, their studies on -S- moieties-containing MOFs for the reduction of proton mainly focused on the enhanced conductivity via S doping as well as S-coordination for optimization behavior of the hydrogen adsorption energy on the metal active center (such as, *Chem. Rev.* 2020, 120, 16, 8536-8580; *ACS Appl. Mater. Interfaces* 2021, 13, 14, 16384-16395; *Angew. Chem. Int. Ed.* 2016, 55, 3566). However, in this work, we used BDMT ligands with sulfhydryl groups for constructing a unique mimic-hydrogenase center (Ni₂-S₁) in the BDC-based MOF, which could efficiently enhance the water activation ability of the center Ni sites and improve the MOF structural stability, which has never been reported previously. Moreover, the newly-developed S-NiBDC catalyst ranks at the top level of HER activity when compared with the representative sulfur-coordinating polymer catalysts (**Table S3**).

As previously reported, the high electrochemical performances for most of the well-developed catalytic electrodes based on metal-dithiolene coordination polymers were tested under acidic conditions, however their HER performances decreased substantially in alkaline media or higher pH electrolyte (*J. Am. Chem. Soc.* 2015, 137, 1, 118-121; *ACS Appl. Mater. Interfaces* 2021, 13, 14, 16384-16395; *Angew. Chem. Int. Ed.*, 54: 12058-12063). In acidic conditions, the HER process is easier than in alkaline conditions due to the direct usage of protons from acidic solution, whereas the protons in alkaline conditions come from the slow water activation. Therefore, the catalytic kinetics of the HER process in alkaline are orders of magnitude slower than in acidic media. Our work mainly focuses on accelerating the water activation kinetics of electrocatalysts for HER under alkaline conditions. The optimal S-NiBDC shows one of the best performances compared with other reported state-of-the-art MOF-based HER electrocatalysts, especially at industrial-level current densities of $>0.5 \text{ A cm}^{-2}$ for practical applications (**Figure S23**). More importantly, the S-NiBDC catalyst even can reach 3.5 A cm^{-2} at a low overpotential of 342 mV and maintain its high catalytic activity over 150 h at 1.0 A cm^{-2} , which is not achieved in the reported MOF-based electrocatalysts (**Figure 3f and 3h**). We believe that

this research direction and conclusion are meaningful for the design and development of high-performance, robust MOF-based HER electrocatalysts.

Table S3. Comparison of electrocatalytic HER performances of S-NiBDC with other reported sulfur-coordinating complex-based HER catalysts.

Catalyst	Electrolyte	Overpotential (mV@mA cm ⁻²)	Tafel slope (mV dec ⁻¹)	Ref.
S-NiBDC	1.0 M KOH	113@10	75	This work
		209@100		
		286@500		
		310@1.0 A cm⁻²		
		328@2.0 A cm⁻²		
342@3.5 A cm⁻²				
CoTHT	pH = 1.3	143@10	70.6	18
THTA-Co/G	0.5 M H ₂ SO ₄	230@10	70	19
CoBHT	pH = 1.3	185@10	88	20
NiBHT		331@10	67	
FeBHT		473@10	119	
CoBHT	pH = 1.3	263@10	149	21
CoTHT		453@10	189	
THTNi 2DSP	0.5 M H ₂ SO ₄	333@10	80.5	22
	0.05 M KOH	574@10	-	
Cu-BHT NP	pH = 0.0	450@10	95	23
NiAT	0.05 M H ₂ SO ₄	370@10	128	24
CoBTT	pH = 1.3	560@10	70	25
NiBTT	pH = 1.3	470@10	76	26
Co-ATTAc ₄ /GR	pH = 1.3	388@10	130	27
Co-PTC	pH = 0.0	227@10	189	28
		610@100		
Cu-MTF	pH = 0.5	96@10	131	29
		463@100		

Figure S23. Comparison of HER performances of S-NiBDC and other reported MOF-based HER catalysts at different current densities.

Figure 3. (f) Chronopotentiometric curves at 1.0 A cm^{-2} for NiBDC and S-NiBDC.

Figure 3. (h) Comparison of achieved highest HER current density and stability time of S-NiBDC and other reported MOF-based and metal/metallic compound catalysts.

Corresponding revisions:

The supporting information part

In response to this comment, we added **Table S3** in supporting information.

Comment 2: *The authors indicate in the opening paragraph that Pt is used in alkaline HER - this is certainly not true. Industrial electrolyzers use NiFe electrodes for proton reduction. The reviewer suggests the authors compare the modified MOF to NiFe to increase the relevance of controls.*

Authors' Response:

We appreciate the valuable comment from the reviewer. Following the reviewer's suggestion, we supplemented the polarization curves of the NiFe electrodes with different metal ratios. As shown in **Figure S56**, the S-NiBDC also displayed higher HER electrocatalytic activity than the NiFe electrodes with different metal ratios, further demonstrating the application potential of S-NiBDC in alkaline industrial electrolyzers. Although the Ni-based electrodes have been acted as the cathode in alkaline industrial electrolyzers due to the scarcity and high cost of Pt electrodes, the Pt-based catalysts are still one of the best groups of state-of-the-art HER electrocatalysts so far (*Nat. Commun.* 2016, 7, 13638; *Nat. Commun.* 2015, 6, 6430). Thus, it is still imperative and challenging to develop the inexpensive and earth-abundant HER electrocatalyst that rivals or even outperforms the Pt-based catalysts, especially in high-current densities of $> 500 \text{ mA cm}^{-2}$.

Figure S56. Polarization curves of S-NiBDC and NiFe foam with different metal ratios.

Corresponding revisions:

The introduction part

In response to this comment, we added **Figure S56** in supporting information and revised the corresponding part on **Page 3** of the revised manuscript as follows:

Expensive and scarce noble metal (e.g., Pt)-based materials are still considered the best electrocatalysts for the hydrogen evolution reaction (HER) involved in water electrolysis⁴.

Comment 3: The authors should normalize their HER rates to ECSA. Are there still differences between the two configurations - BDC only vs. mixed BDC/BDMT. The reviewer imagines so, however, it would be good to see in the SI.

Authors' Response:

We appreciate the valuable comment from the reviewer. Yes, the polarization curves normalized by ECSA are effective to compare the intrinsic activities of HER catalysts. Following the reviewer's suggestions, we supplemented the analysis of ECSA-normalized polarization curves. As we know, the ECSA of one material with similar composition is proportional to its electrochemical double-layer capacitance (C_{dl}), which could be measured by CV curves in a non-Faradaic region at different scan rates (**Figure S18**). The obtained data are further fitted to get the value of C_{dl} . In **Figure S19**, the S-NiBDC showed a higher C_{dl} value of 15.1 mF cm^{-2} than the NiBDC (4.4 mF cm^{-2}), suggesting more exposed active sites of S-NiBDC. The ECSA of each sample can be evaluated from C_{dl} according to the following equation:

$$\text{ECSA} = C_{dl} / C_s$$

where C_s is the specific capacitance of the sample or the capacitance of an atomically smooth planar surface of the material per unit area under identical electrolyte conditions. The C_s is usually found to be in the range of 0.02-0.06 mF cm^{-2} , and it is assumed as 0.04 mF cm^{-2} in the calculations of ECSA (*J. Am. Chem. Soc.* 2013, 135, 45, 16977-16987; *Angew. Chem. Int. Ed.* 2019, 58, 4679-4684). The polarization curves of samples normalized to ECSA are re-plotted in **Figure S20**, and the results further confirm that the HER activity of S-NiBDC does increase intrinsically compared with that of bare NiBDC.

Figure S18. (a-b) CV curves of S-NiBDC and NiBDC at different scan rates in 1.0 M KOH.

Figure S19. ECSAs of NiBDC and S-NiBDC.

Figure S20. Polarization curves of NiBDC and S-NiBDC normalized by ECSA.

Corresponding revisions:

In response to this comment, we added **Figure S20** in supporting information and revised the corresponding part on **Page 8** of the revised manuscript as follows:

1) The results and discussion part

In addition, the comparing polarization curves normalized by ECSA for S-NiBDC and NiBDC verify the enhanced intrinsic HER activity (Figure S20)³⁹.

2) The supporting information part

The ECSA of each sample can be evaluated from C_{dl} according to the following equation:

$$ECSA = C_{dl} / C_s$$

where C_s is the specific capacitance of the sample or the capacitance of an atomically smooth planar surface of the material per unit area under identical electrolyte conditions. The C_s is usually found to be in the range of 0.02-0.06 mF cm⁻², and it is assumed as 0.04 mF cm⁻² in the calculations of ECSA.

Comment 4: The stability of the mixed NiBDC/BDMT material is questionable. It is certainly more stable than BDC, but there is a clear loss of crystallinity in the PXRD for the mixed MOF. Beyond PXRD and SEM, was any additional post-catalysis characterization completed? Are there dissolved species in the electrolysis solution (NMR or ICP-MS)? Is the distribution of Ni species unchanged (XPS)?

Authors' Response:

We appreciate the valuable comment from the reviewer. Following the reviewer's suggestion, we firstly measured XPS and XAS characterizations of S-NiBDC after HER electrocatalysis to figure out the situation in valence states and coordination environment of center Ni species. As shown in **Figure S28a and S29a**, it is clearly seen that the oxidation states of Ni species are hardly changed after HER catalytic reaction based on XPS and Ni K-edge XANES results; also the Ni-S bond centered at 162.0 eV could still be observed in the S-NiBDC (**Figure S28b**). Besides, it can be found that in the quantitative EXAFS fitting analyses, the first shell coordination for S-NiBDC after HER reaction is identical to the pristine sample (**Figure S29b, S30 and Table S1**), highlighting the structural stability of S-NiBDC. Secondly, we conducted $^1\text{H-NMR}$ spectra of the electrolytes after HER reaction for both NiBDC and S-NiBDC. As displayed in **Figure S26**, as for NiBDC, the $^1\text{H-NMR}$ spectrum of the electrolyte after HER shows the release of the BDC organic linkers because of its structural destruction. However, no detected dissolved organic species come from the BDC-based MOF structure in the electrolyte of S-NiBDC after HER (*Angew. Chem. Int. Ed.* 2016, 55, 16049), further proving the structural stability of S-NiBDC. Due to the species loss on the surface in strong alkali condition (*ACS Energy Lett.* 2018, 3, 6, 1360-1365; *Nat. Energy* 2020, 5, 881-890), it is believed that the loss of crystallinity in the XRD pattern for the post-catalysis S-NiBDC sample is inevitable. However, it is worth noting that such a loss did not affect the valence states and the enhanced structural stability of S-NiBDC, based on the above supplemented post-catalysis experiments.

Figure S28. High-resolution XPS spectra for (a) Ni 2p and (b) S 2p of S-NiBDC after HER tests.

Figure S29. (a) Ni K -edge XANES and (d) k^3 -weighted EXAFS spectra of S-NiBDC before and after HER tests.

Figure S30. Fourier transform-EXAFS fitting results of S-NiBDC after HER tests.

Table S1. EXAFS fitting parameters at the Ni K -edge for various samples ($S_0^2=0.92$).

Sample	Shell	CN^a	$R(\text{Å})^b$	$\sigma^2(\text{Å}^2)^c$	$\Delta E_0(\text{eV})^d$	R factor (%)
Ni K-edge						
Ni foil	Ni-Ni	12*	2.47 ± 0.003	0.0047	5.76 ± 0.43	0.3
NiBDC	Ni-O	5.6 ± 0.2	2.03 ± 0.013	0.0091	3.62 ± 1.46	0.7
S-NiBDC	Ni-O	5.1 ± 0.2	1.99 ± 0.022	0.0089	6.41 ± 3.38	0.8
	Ni-S	0.9 ± 0.1	2.15 ± 0.010	0.0076	5.02 ± 2.92	
S-NiBDC-after HER	Ni-O	5.5 ± 0.1	2.08 ± 0.052	0.0096	5.06 ± 5.40	0.3
	Ni-S	0.8 ± 0.3	2.14 ± 0.033	0.0048	4.11 ± 2.37	

Figure S26. ^1H -NMR spectra of the electrolytes after HER tests for NiBDC and S-NiBDC.

Corresponding revisions:

The results and discussion part

In response to this comment, we added **Figure S26, S28-30** and updated **Table S1** in supporting information, and revised the corresponding part on **Page 9-10** of the revised manuscript as follows:

The enhanced structural stability of S-NiBDC is further evidenced by XRD results, which display a limited change in the crystal structure of S-NiBDC after HER tests; in contrast, significant degradation is observed for NiBDC (Figure S24-S25).

Further, no morphological change was observed by FESEM imaging when progressively surging the current density even up to 1.0 A cm^{-2} , while no dissolved organic species in the electrolyte of S-NiBDC after HER were detected⁴⁵ (Figure 4e and Figure S26).

Post XPS and XAS analysis on S-NiBDC further reveal hardly changed oxidation states and coordination situations (Figure S28-S30).

Comment 5: *The reviewer appreciates the deep mechanistic discussion, including the characterization of the S-H* intermediate. This result is interesting, although potentially expected given previous studies with dithiolene compounds, where the protonation of the S or N linkage is preferred to metal binding. That said, how does the S-H bond formation impact the local geometry within the MOF. This comment refers back to the stability paragraph - in previous non-innocent ligand arrangements protonation at a S or N resulted in bond breakage between the node and linker, and thus, a pathway for instability.*

Authors' Response:

Thanks for the valuable comment from the reviewer. Following the reviewer's suggestion, we supplemented the structural diagrams of S-NiBDC in the HER process based on theoretical calculations. From

these optimized geometries, it is found that the S-H bonds can be formed without breaking other chemical bonds, even without influencing the geometry of the localized “Ni₂-S₁” active region (**Figure S41**). That is to say, during the HER process, the protonation of S-NiBDC at the S sites does not result in bond breakage between the node and linker. The result is further supported by experimental observations. In **Figure S26**, the ¹H-NMR characterization of the electrolyte after the long-term HER test shows that there are no dissolved organic linkers observed in the solution for the S-NiBDC sample. In addition, the post- XPS and XAS characterizations of S-NiBDC after HER reaction also show that the valence states and coordination environments of the Ni species were not changed significantly (**Figure S28-S30**). All the above experimental results demonstrate that the S-H bond could be stable formed in the S-NiBDC model.

Figure S41. Schematic illustration of the HER process for S-NiBDC in alkaline solution.

Figure S26. ¹H-NMR spectra of the electrolytes after HER tests for NiBDC and S-NiBDC.

Figure S28. High-resolution XPS spectra for (a) Ni 2p and (b) S 2p of S-NiBDC after HER tests.

Figure S29. (a) Ni K -edge XANES and (d) k^3 -weighted EXAFS spectra of S-NiBDC before and after HER tests.

Figure S30. Fourier transform-EXAFS fitting results of S-NiBDC after HER tests.

Corresponding revisions:

The supporting information part

In response to this comment, we added **Figure S26, S28-S30, and S41** in supporting information.

Comment 6: *The characterization of the MOFs themselves seems relatively standard. The PXRD patterns do seem to exhibit a high background. Could a zero-background holder be used to eliminate this noise?*

Authors' Response:

We appreciate the valuable comment from the reviewer. Following the reviewer's suggestion, we re-measured the PXRD patterns of NiBDC and S-NiBDC samples without growing on the substrate to realize a zero background. As shown in **Figure S7**, the S-NiBDC displays typical BDC-based MOF characteristic peaks that are similar to the NiBDC, suggesting that the original crystal structure is still maintained and no new phase is formed after BDMT doping.

Figure S7. XRD patterns of S-NiBDC and NiBDC without substrates.

Corresponding revisions:

The supporting information part

In response to this comment, we added **Figure S7** in supporting information.

Thanks again for your valuable comments to help us improve the quality of our manuscript!

Response to the comments of Reviewer #3

Comments: *The authors of this paper report the fabrication of S-doping Ni-benzenedicarboxyl acid (BDC)-based metal-organic framework (S-NiBDC) as a hydrogen evolution (HER) electrocatalyst in alkaline solution. By theoretical calculations and various structural characterizations, it is claimed that the triangular “Ni₂-S₁” motif in S-NiBDC catalysts could facilitate the water activation kinetics for boosting high HER activity. However, the concept of dopant-regulated BDC-based catalysts toward improved HER performance has been reported before. Moreover, the synthetic approach of S-NiBDC catalysts in this work seems very routine and their structures are questionable. Based on these considerations, this reviewer fails to recommend the publication of this manuscript in Nature Communications. I hope the following comments will be helpful for improving the quality of this manuscript.*

Authors' Response:

We appreciate the valuable comment from the reviewer.

Although the doping-regulation of BDC-based MOF materials toward improved HER electrocatalytic performance have been reported previously, it has mainly focused on modulating the electronic structure of the metal active sites (*Nat. Mater.* 2022, 21, 673-680; *Nat. Commun.* 2021, 12, 1369; *Adv. Mater.* 2021, 33, 2106781; *Adv. Energy Mater.* 2018, 8, 1801065; *Nat. Commun.* 2017, 8, 15341). Instead of well-developed doping modulation, in this work, we propose a unique ligand modulation strategy to construct the unique triangular active-regions that is the ternary active structure with “Ni₂-S₁” mimic-hydrogenase motifs. Further, we clearly confirm, based on electrochemical results combined with advanced structural characterizations, that the constructed mimic active region and the enhanced HER electrocatalytic mechanism. This concept has never been reported previously. More importantly, this concept is highly general, which is used to develop three different S-MBDC catalysts (e.g., S-CoBDC, S-FeBDC, S-NiBDC) with triangular M₂-S₁ active regions, and the corresponding alkaline HER activities of CoBDC, FeBDC, or NiBDC can indeed be improved by introducing “Co₂-S₁”, “Fe₂-S₁”, “Ni₂-S₁” active regions.

Moreover, this work mainly focuses on accelerating the water activation kinetics of electrocatalysts for HER under alkaline conditions. The optimal S-NiBDC shows one of the best performance compared with other reported state-of-the-art MOF-based HER electrocatalysts, especially at industrial-level current densities of > 0.5 A cm⁻² for practical applications. More importantly, the S-NiBDC catalyst even can reach 3.5 A cm⁻² at a low overpotential of 342 mV and maintain its high catalytic activity over 150 h at 1.0 A cm⁻², which is not achieved on the reported MOF-based electrocatalysts (**Figure 3f and 3h**). We believe that this research direction and conclusion are meaningful for the design and development of high-performance, robust MOF-based HER electrocatalysts.

Figure 3. (f) Chronopotentiometric curves at 1.0 A cm^{-2} for NiBDC and S-NiBDC.

Figure 3. (h) Comparison of achieved highest HER current density and stability time of S-NiBDC and other reported MOF-based and metal/metallic compound catalysts.

Comment 1: *The proposed structural model of S-NiBDC in Figure 1b is not consistent well with the practical catalysts. There should be the coexistence of benzenedicarboxyl and 1,4-benzenedimethanethiol ligands in the S-NiBDC catalysts, rather than the presence of only 1,4-benzenedimethanethiol linkers. Moreover, in the S-NiBDC model, partial S atoms bind with two Ni atoms and the others bind with only one Ni atom. Why?*

Authors' Response:

We appreciate the valuable comment from the reviewer. Following the reviewer's suggestion, we re-drew the models of NiBDC and S-NiBDC. As shown in **Figure 1b**, the proposed S-NiBDC model is built up by 1,4-dicarboxybenzene acid (BDC) and 1,4-benzenedimethanethiol (BDMT) organic linkers, which is consistent with practical catalysts. In the previous version, we chose an incorrect placement angle when drawing the model before, and the ternary structure of two Ni atoms binding with the S atom ($\text{Ni}_2\text{-S}_1$) in the

lower part could not be observed clearly. Therefore, in the revised version, we chose the appropriate angle so that we could clearly observe the Ni₂-S₁ motifs formed in S-NiBDC when we redrew the model this time.

Figure 1. Models of optimized NiBDC (a) and S-NiBDC (b) structures.

Corresponding revisions:

The results and discussion part

In response to this comment, we updated **Figure 1a-b** in the revised manuscript.

Comment 2: *Based on the p band center of S-NiBDC shown in Figure 1d, it is claimed that S-NiBDC possesses beneficial properties for HER. But the p band center of S-NiBDC is located at -2.94 eV, which is still very far from the Fermi level.*

Authors' Response:

We appreciate the valuable comment from the reviewer. Yes, we also noticed this problem and recognized that the statements should be made more clear for the p -band. Following the reviewer's suggestion, the following presentation about the p band states has been added in the manuscript as follows.

On Page 5: "After the introduction of S species, clearly, the p -band center (ϵ_p) of S-NiBDC model has been pushed closer to the Fermi level than that of NiBDC model with an increase of 2.53 eV, which indicates that the p states of S-NiBDC structure contribute to the high HER activity more easily (**Figure 1d**)²⁶. Besides, the introduced S species bring more states for the bonding and anti-bonding orbitals, which could help to the intermediates adsorption and electron transport for the whole HER reaction."

On Page 12: "As displayed in **Figure 6d**, the introduced S sites of S-NiBDC binding with H atom show that the total charge transport capacity of the system is enhanced, and the spin-up and spin-down are closer to the Fermi level. The spin-down demonstrates more states on the Fermi level, which indicates the S site in S-NiBDC model has more hybridization with the H atom (**Figure S38-S39**) than the Ni site. Meanwhile, this hybridization brings the better electron transport performance for the catalyst and reaction process which could be one of the reasons to boost the HER."

Figure 1. (d) PDOS of p -states for NiBDC and S-NiBDC models.

Figure 6. (d) PDOS of different sites for S-NiBDC with H binding.

Figure S38. PDOS of p - and d - states of different sites for S-NiBDC with H binding.

Figure S39. PDOS of H and S atom.

Corresponding revisions:

In response to this comment, we added **Figure 6d** and **Figure S38-S39** and revised the corresponding part on **Page 5** and **Page 12** in the revised manuscript.

Comment 3: *The average metal oxidation states of Ni species in both NiBDC and S-NiBDC catalysts are recommended to be given, based on the results of XANES and XPS spectra.*

Authors' Response:

We appreciate the valuable comment from the reviewer. Following the reviewer's suggestion, we provided the oxidation states of Ni species in NiBDC and S-NiBDC catalysts based on the calibration curve of chemical valences with the Ni *K*-edge energy position from the first derivative of XANES. The Ni foil and NiO references are taken from the standard testing. The chemical value for LaNiO₃ (III) standard reference is taken from the literature (*J. Mater. Chem.*, 2011,21, 18592-18596). The Ni *K*-edge energy position correlates proportionally nonlinearity with the formal valences of the reference samples (*Nat. Energy* 2020, 5, 881-890). As shown in **Figure S11**, the average oxidation state of Ni species in the NiBDC is +1.9, which is less than the standard Ni +2 valence due to the partial unsaturated sites generated during synthesis (*Adv. Mater.* 2020, 32, 2006784). In contrast, the oxidation state of Ni species for S-NiBDC is +2.2, which is higher than that of control NiBDC, suggesting the coordination modulation by S species adding.

Figure S11. The oxidation states of Ni species plotted against Ni *K*-edge position obtained from the first derivative method.

Corresponding revisions:

In response to this comment, we added **Figure S11** in supporting information and revised the corresponding part on **Page 7** of the revised manuscript as follows:

1) The results and discussion part

In the Ni *K*-edge X-ray absorption near-edge structure (XANES) spectra (Figure 2c and Figure S11), the oxidation state of Ni species in the S-NiBDC increases to +2.2 with respect to that of NiBDC (+1.9), indicating an increased valence state of Ni species, in consistence well with the XPS results³⁰.

2) The supporting information part

The Ni foil and NiO references are taken from the standard testing. The chemical value for LaNiO₃ (III) standard reference is taken from the literature (J. Mater. Chem., 2011, 21, 18592-18596). The Ni *K*-edge energy position correlates nonlinearly with the formal valences of the reference samples.

Comment 4: *The XAFS fitting results shown in Table S1 can not clearly support the formation of triangular “Ni₂-S₁” region in the S-NiBDC catalysts. Is there any evidence to confirm this point?*

Authors’ Response:

We appreciate the valuable comment from the reviewer. Firstly, the results of high-resolution S 2p XPS spectra and S *K*-edge XANES spectra of S-NiBDC confirmed that the doping of the BDMT ligand led to the successful formation of the Ni-S bond. Secondly, the XAFS fitting results in **Table S1** revealed the formation of Ni-O₅S₁ coordination for S-NiBDC by breaking the original symmetrical Ni-O₆ coordination after S species introduction. Thirdly, as revealed by DFT calculations, the original two O atoms in BDC were replaced by one S atom in BDMT, forming the ternary “Ni₂-S₁” region by combining with two adjacent Ni atoms in Ni-oxide layers (**Figure 1**). Moreover, the stability of the coordination structure and valence state was also demonstrated

by the supplemented characterization analysis and the schematic diagram of the theoretical HER process after long-term reaction (Figure S28-S30 and S41).

Table S1. EXAFS fitting parameters at the Ni K-edge for various samples ($S_0^2=0.92$).

Sample	Shell	CN ^a	R(Å) ^b	$\sigma^2(\text{Å}^2)$ ^c	$\Delta E_0(\text{eV})$ ^d	R factor (%)
Ni K-edge						
Ni foil	Ni-Ni	12*	2.47±0.003	0.0047	5.76±0.43	0.3
NiBDC	Ni-O	5.6±0.2	2.03±0.013	0.0091	3.62±1.46	0.7
S-NiBDC	Ni-O	5.1±0.2	1.99±0.022	0.0089	6.41±3.38	0.8
	Ni-S	0.9±0.1	2.15±0.010	0.0076	5.02±2.92	
S-NiBDC-after HER	Ni-O	5.5±0.1	2.08±0.052	0.0096	5.06±5.40	0.3
	Ni-S	0.8±0.3	2.14±0.033	0.0048	4.11±2.37	

Figure 1. Models of optimized NiBDC (a) and S-NiBDC (b) structures.

Figure S28. High-resolution XPS spectra for (a) Ni 2p and (b) S 2p of S-NiBDC after HER tests.

Figure S29. (a) Ni K -edge XANES and (d) k^3 -weighted EXAFS spectra of S-NiBDC before and after HER tests.

Figure S30. Fourier transform-EXAFS fitting results of S-NiBDC after HER tests.

Figure S41. Schematic illustration of the HER process for S-NiBDC in alkaline solution.

We further checked EDS spectra for both samples, and the results confirmed the S species formed in S-NiBDC, while no S species existed in NiBDC (**Figure X1**).

Figure X1. EDS spectra for S-NiBDC and NiBDC

Fourthly, additional AC-STEM observation for S-NiBDC were performed to directly probe into atomic-level arrangements of S and Ni species in the S-NiBDC. We simultaneously obtained high-angle annular dark-field scanning transmission electron microscopy (HAADF-STEM) image and integrated differential phase contrast (iDPC) image for S-NiBDC. Since the intensity of HAADF-STEM image is roughly proportional to Z^2 (Z is the atomic number) and iDPC image intensity also correlates to Z , the brighter contrast in both types of images comes from Ni or S atoms. As given below, obvious intensity can be observed between Ni layers, indicating S atoms reside between them. Especially, the iDPC image likely evidences our proposed triangular “Ni₂-S₁” model with longer and shorter Ni-Ni distances (**Figure X2**). Therefore, based on the above XPS, XANES, XAFS fitting results, AC-STEM observation, and DFT calculations, one can conclude that the triangular “Ni₂-S₁” active region motifs were successfully constructed in the BDC-based MOF structure.

Figure X2. The HAADF-STEM and iDPC images for S-NiBDC.

Comment 5: Moreover, the WT-EXAFS plots of NiBDC and S-NiBDC catalysts look similar to each other (Figure 2e). Since the Ni-S bonds are claimed to form in S-NiBDC catalyst, there should be some difference between the WT-EXAFS plots of NiBDC and S-NiBDC samples.

Authors' Response:

Thanks for the valuable comment from the reviewer. Following the reviewer's suggestion, we re-arranged the WT-EXAFS plots to clearly discern the Ni *K*-edge EXAFS oscillations and offered the higher resolution in both *k* and *R* space, which is widely used for investigating the coordination environment of Ni species in the various samples (*Angew. Chem. Int. Ed.* 2022, 61, e202117347). As shown in **Figure 2e** and **Figure S13**, the WT contour plot of S-NiBDC exhibits a maximum peak at around 3.9 Å⁻¹ derived from the main coordinated O atoms, and the peak located at 6.1 Å⁻¹ can be attributed to the Ni-S coordination that doesn't exist in the NiBDC. This result demonstrates the predominantly O-coordinated BDC-based MOF structure with introduced S atoms as partial coordinators (*Nat. Commun.* 2020, 11, 3049; *Small* 2022, 18, 2105387). The different local environments of center Ni species between S-NiBDC and NiBDC can be attributed to the contribution of the backscattering between Ni and the different light O/S atoms (*Nat. Commun.* 2021, 12, 6766; *Sci. Adv.* 2020, 6, eaba6586).

Figure 2. (e) WT-EXAFS spectra of NiBDC, S-NiBDC, NiO, and NiS.

Figure S13. WT-EXAFS spectrum of Ni foil.

Corresponding revisions:

The results and discussion part

In response to this comment, we updated **Figure 2e** and **Figure S13** and revised the corresponding part on **Page 7** of the revised manuscript as follows:

Further analyses on the wavelet transform (WT) (Figure 2e and Figure S13) show that the maximum intensity of S-NiBDC closed to 3.9 \AA^{-1} is derived from the light O atoms coordination, while the peak located at 6.1 \AA^{-1} can be attributed to the Ni-S bonds, indicating the predominantly O-coordinated BDC-based MOF structure with introduced S atoms as partial coordinators^{33,34}.

Comment 6: *In general, the pure NiBDC catalyst exhibits poor electrochemical HER performance. However, in this work, the HER activity of NiBDC sample is superior to that of the benchmark Pt/C catalysts. What is the reason for this?*

Authors' Response:

We appreciate the valuable comment from the reviewer. Following the reviewer's suggestion, we provided and compared the polarization curves for NiBDC, S-NiBDC, and Pt/C samples. As shown in **Figure S16**, at the current densities of below 400 mA cm^{-2} , actually, the NiBDC requires the higher overpotentials than the commercial Pt/C catalyst to reach the same current density, indicating the lower HER performance of NiBDC than that of commercial Pt/C at low current density. However, at higher current densities up to 0.5 and 1.0 A cm^{-2} , the HER catalytic activity of commercial Pt/C is significantly lower than that of NiBDC (**Figure 3c**), which may be mainly attributed to the different structure of the electrode. In the preparation of electrode materials, the control NiBDC array was in situ grown on the surface of 3D conductive Ni foam (NF) (**Figure S15**), while commercial Pt/C was drop cast on the surface of 3D NF with the same loading amount. Thanks to the unique nanoarray structure rather than the traditional powder sample, the HER catalytic activity of as-prepared NiBDC catalyst can exceed that of commercial Pt/C catalyst at high current densities (*Energy Environ. Sci.*, 2020, 13, 3007-3013; *J. Am. Chem. Soc.* 2021, 143, 23, 8720-8730; *J. Am. Chem. Soc.* 2022, 144, 13, 6028-6039; *Appl. Catal. B* 2022, 305, 121081).

Figure S16. Polarization curves of NiBDC, S-NiBDC, and Pt/C.

Figure 3. (c) Overpotentials at 0.5 and 1.0 A cm⁻² of NiBDC, S-NiBDC, and Pt/C.

Figure S15. (a-b) FESEM images of NiBDC.

Corresponding revisions:

The supporting information part

In response to this comment, we added **Figure S16** in supporting information.

Comment 7: To solidly demonstrate the structural stability, XAFS spectroscopy is suggested to be performed for the S-NiBDC catalysts after a long-term HER operation in alkaline.

Authors' Response:

Thanks for the valuable comment from the reviewer. Following the reviewer's suggestion, we conducted additional XAS characterization of S-NiBDC after long-term HER electrocatalysis in alkaline. As shown in **Figure S29a**, essentially no changes were observed in the Ni *K*-edge XANES spectra for S-NiBDC before and after HER reaction, demonstrating the excellent structural stability of S-NiBDC. In addition, the coordination environment of center Ni species was further determined by EXAFS and the quantitative fitting analyses (**Figure S29b, S30, and Table S1**). Clearly, the large major peak at around 1.57 Å corresponding to the combined scatterings of Ni-O and Ni-S bonds still existed; after HER reaction, the position of the major peak on S-NiBDC was slightly larger than that of the pristine catalyst before HER (1.56 Å), due to the fact that partial surface oxidation is inevitable for post-catalysis. Based on the above results, overall, the S-NiBDC catalyst is proved to be highly stable after long-term HER operation.

Figure S29. (a) Ni *K*-edge XANES and (d) *k*³-weighted EXAFS spectra of S-NiBDC before and after HER tests.

Figure S30. Fourier transform-EXAFS fitting results of S-NiBDC after HER tests.

Table S1. EXAFS fitting parameters at the Ni K-edge for various samples ($S_0^2=0.92$).

Sample	Shell	CN ^a	R(Å) ^b	$\sigma^2(\text{Å}^2)^c$	$\Delta E_0(\text{eV})^d$	R factor (%)
Ni K-edge						
Ni foil	Ni-Ni	12*	2.47±0.003	0.0047	5.76±0.43	0.3
NiBDC	Ni-O	5.6±0.2	2.03±0.013	0.0091	3.62±1.46	0.7
S-NiBDC	Ni-O	5.1±0.2	1.99±0.022	0.0089	6.41±3.38	0.8
	Ni-S	0.9±0.1	2.15±0.010	0.0076	5.02±2.92	
S-NiBDC-after HER	Ni-O	5.5±0.1	2.08±0.052	0.0096	5.06±5.40	0.3
	Ni-S	0.8±0.3	2.14±0.033	0.0048	4.11±2.37	

Corresponding revisions:**The results and discussion part**

In response to this comment, we added **Figure S29-S30**, updated **Table S1** in supporting information, and revised the corresponding part on **Page 10** of the revised manuscript as follows:

Post XPS and XAS analysis on S-NiBDC further reveal hardly changed oxidation states and coordination situations (Figure S28-S30).

Comment 8: As the in-situ Raman spectra shown in Figure 5c, the vibrational signal of S-H* bonds can be obtained till the potentials of -1.0 V or -1.2 V is applied. It is claimed that the S-NiBDC catalyst deliver high current density of 1 A cm⁻² at a relatively low overpotential of 310 mV. Why the vibrational signal of S-H* bond is undetectable at a low potential, such as -0.4 or -0.5 V?

Authors' Response:

We appreciate the valuable comment from the reviewer. In the previous version, the inconsistent Raman signal may be attributed to the fact that surface-enhanced Raman spectroscopy that requires Au or Ag as a substrate to enhance the Raman signal from chemical species close to the surface was not used (*Angew. Chem. Int. Ed.* 2021, 60, 19774). As a result, the peak of S-H* intermediate by increasing the applied potential window and the corresponding tested HER potential range are not a good match. Thus, the results obtained from the Raman test could not fully correspond to the potential window of the HER test, which allowed for the observation of the S-H* signal only at lower voltages (> -0.8 V vs. RHE) (*Angew. Chem. Int. Ed.* 2021, 60, 19774). In order to solve the above problems, we supplemented the in-situ attenuated total reflectance infrared absorption spectroscopy (ATR-FTIR) for S-NiBDC and NiBDC samples with the ultra-thin Au foil deposited on the silicon to enhance the signal of S-H* intermediate. **Figure 5c-d** showed the spectra variation of both S-NiBDC and NiBDC samples after applying the potentials from 0 to -0.4 V vs. RHE. As shown in **Figure 5c**,

an absorption peak located at 2555 cm^{-1} was observed for S-NiBDC at -0.2 V vs. RHE and gradually strengthens in intensity with the applied potential increased to -0.4 V vs. RHE , which can be attributed to the formed S-H* intermediates during the HER process (*Angew. Chem. Int. Ed.* 2021, 60, 7251). However, as for NiBDC, no S-H* vibration absorption can be found at the same applied potential window (**Figure 5d**). These consequences reveal that the S species in S-NiBDC is the catalytic active sites and acts as H atom catcher for further Heyvosky step to produce H_2 .

Figure 5. (c-d) *In-situ* ATR-FTIR spectra of S-NiBDC and NiBDC with the potential of 0 to -0.4 V vs. RHE .

Corresponding revisions:

The results and discussion part

In response to this comment, we updated **Figure 5c-d** and revised the corresponding part on **Page 10** of the revised manuscript as follows:

To further identify the definite role of S species, *in-situ* attenuated total reflectance infrared absorption spectroscopy (ATR-FTIR) measurements were recorded. As shown in Figure 5c, an absorption peak located at 2555 cm^{-1} was detected and gradually strengthened on S-NiBDC from 0 to -0.4 V vs. RHE , which was ascribed to the S-H* intermediate⁴⁷. In contrast, the vibration absorption band of S-H* intermediate was not found for NiBDC in the same potential window (Figure 5d). These results demonstrate that the key S-H* intermediate was only formed on the S-NiBDC during the HER process in alkaline conditions, indicating the introduction of S active species acted as H atom catchers for facilitating the H_2 evolution.

Comment 9: Based on the Gibbs free-energy calculations in Figure 6d, the formation of *H on the S site of S-NiBDC model has the lowest potential barrier of 1.19 eV, which is still much larger than the optimal value

for HER. The Gibbs free energy change of Pt catalysts during HER is suggested to be provided here as reference.

Authors' Response:

We appreciate the valuable comment from the reviewer. Based on the calculations, we have fixed problems and provided the new results in Figure 6c. From the figure, the formation of *H on the S site of S-NiBDC model should be 0.35 eV. Moreover, following the reviewer's suggestion, we provided the Gibbs free energy change of Pt catalysts during HER as reference, and the results show that the Gibbs free energy change of Pt (111) during HER is about -0.09 eV, which has been provided in the manuscript.

Corresponding revisions:

The supplemented references (Page 10)

52 Nørskov, J. K. et al. Trends in the Exchange Current for Hydrogen Evolution. J. Electrochem. Soc. 152, J23, (2005).

Thanks again for your valuable comments to help us improve the quality of our manuscript!

REVIEWER COMMENTS

Reviewer #1 (Remarks to the Author):

The authors have mostly responded to my previously raised concerns, while there are a few questions to be addressed before publication in Nature Communications.

1. The reviewer still concerns the quality of DFT calculations due to:

1) The data of water adsorption energy and hydrogen adsorption energy in Figure 6 is quite different from the previous version while no explanations were provided.

2) The free energy changes of *H adsorption in NiBDC is surprisingly high (1.20 eV), which is inconsistent with its relative high activities, e.g., no substantial difference on intrinsic activity compared to S-NiBDC (only 0.35 eV free energy changes of *H adsorption) and even better HER performance than Pt in Figure 3c.

2. In Figure 3f, the stability test of NiBDC should also be conducted in a refreshed electrolyte similar to the S-NiBDC for a better comparison.

3. In line 220 page 12, it is strange to demonstrate “the vibration absorption band of S-H* intermediate was not found for NiBDC”, because there is even no S element in the NiBDC sample.

Reviewer #4 (Remarks to the Author):

This revised manuscript proposed S-doped NiBDC nanosheet arrays with the “Ni₂-S₁” active centers through a universal ligand regulation strategy. The electrocatalyst S-NiBDC was capable of reaching the current density of 1.0 A cm⁻² and maintaining stability for more than 150 h, which can be further increased to 3.5 A cm⁻². This electrocatalytic performance of S-NiBDC presents a breakthrough in noble-metal-free MOF-based HER electrocatalysts and makes a significant contribution to the research in this field. Detailed characterizations demonstrated the successful construction of the “Ni₂-S₁” mimic-hydrogenase active regions in the S-NiBDC structure. Moreover, the Operando EIS measurements, KIE tests, in-situ ATR-FTIR spectra, and DFT calculations confirmed the accelerated water activation kinetics and deeply revealed the electrocatalytic mechanism of HER at the constructed triangular “Ni₂-S₁” sites. The superior stability of S-NiBDC with a ternary active site compared to NiBDC was also verified.

After carefully reading it, I find this work very interesting and stand high priority compared to recently reported MOF-based water splitting catalysts. The whole concept is well-established, the experiments are also very well done, the findings show good novelty, and the mechanism and in-situ studies provide an insightful understanding of alkaline HER on the Ni-based MOF. More importantly, the authors have responded well and thoroughly to the previous reviewers' comments; I think all the queries have been adequately addressed and well explained. Therefore, I would like to suggest that this significantly revised manuscript can be completely recommended for publication in Nature Communications. But before the full acceptance, I would like to suggest the authors further polish the manuscript via the following minor amendments.

1. Figure 3f shows that the performance of NiBDC decreases rapidly during HER; what are the reasons for its performance degradation? Some explanation is needed.
2. The result of EIS in Figure S17 showed obvious differences among the samples; could you provide the specific values?
3. Figure 6e, the schematic illustration of the accelerated water activation for HER in S-NiBDC (right) and NiBDC (left) is too cartoon; I suggest the authors try to draw a new one from the molecule or atom level rather than the bulk materials, which will make the scheme more insightful to the readers.

Response to Reviewers' Comments

Itemized Response to the Comments on Nature Communications (Manuscript number: NCOMMS-22-14896A-Z).

Title: Accelerated Water Activation and Stabilized Metal-Organic Framework via Constructing Triangular Active-Regions for Ampere-Level Current Density Hydrogen Production

For your reference, the detailed point-by-point response is listed below. To save your and the reviewers' time to quickly read what we have done for the revision, in this response letter, the texts for “**Authors' Response**” are marked in blue font, and the changes are highlighted in **YELLOW** color in both revised manuscript and Supporting Information.

Response to the comments of Reviewer #1

Comments: *The authors have mostly responded to my previously raised concerns, while there are a few questions to be addressed before publication in Nature Communications.*

Authors' Response:

Thank you very much for all of the comments on our manuscript. As shown in the following responses, we have supplemented more detailed and systematic data in the revised manuscript to make our work more solid. We hope that the reviewer finds our manuscript suitable for the publication in Nature Communications.

Comment 1: *The reviewer still concerns the quality of DFT calculations due to:*

- 1) The data of water adsorption energy and hydrogen adsorption energy in Figure 6 is quite different from the previous version while no explanations were provided.*
- 2) The free energy changes of *H adsorption in NiBDC is surprisingly high (1.20 eV), which is inconsistent with its relative high activities, e.g., no substantial difference on intrinsic activity compared to S-NiBDC (only 0.35 eV free energy changes of *H adsorption) and even better HER performance than Pt in Figure 3c.*

Authors' Response:

We appreciate the valuable comment from the reviewer.

1) In the first revision, we thought that the problem of high adsorption free energies should be attributed to the modified value about the reaction environment (pH = 14). That is, the ΔG_{pH} can be considered as stated in Equation S6 of SI and it goes to 0.8288 eV ($\Delta G_{\text{pH}} = \text{pH} \times \text{kBTln}10$) for the calculation of reaction free energy in this work. After having checked with many important references, we found this value can be worn off

without affecting the intrinsic mechanism and whole trend of the reaction basically. Therefore, we used the data after deducting this modified value, which is the reason why the data in the first revision is different from the initial version.

2) After the previous revision made, we kept thinking about what the problem existing and how to get an improved method for reaction simulation. Meanwhile, we also checked two important references (*Nat. Commun.* 2021, 12, 1369; *Angew. Chem. Int. Ed.* 2021, 60, 22276) and found that the similar systems were used to simulate the same reaction, but different results about the relative free energies were obtained. Thus, we submitted the same scripts to the server to repeat the whole process. After several days of energy calculation, the process went to the frequency step of the phonon calculation. The k -point of the system has been increased to $3*3*3$ to avoid the influence of imaginary frequency. For the new results, we have tried to consider the ΔG_{pH} and even added the pH correction value to the H energy calculation ($E(H) = 1/2H_2 - 0.0592 \times pH$). At this time, the same adsorption energies with different zero point energies were obtained (**Figure 6b-c**). In details, the Ni sites in the S-NiBDC still exhibited the highest water adsorption ability and the S site delivered the optimal ΔG_{H^*} ; these values are different from the first revision. Now, the obtained H^* adsorption value of -0.38 eV in the NiBDC is more reasonable and is close to the values of the identical calculated systems reported in the literature. Revision has been made accordingly.

To ensure the accuracy of the calculations, we repeated the process on our server and even submitted it to another server. The results obtained from multiple calculations are consistent. For the previous problem, we carefully checked the server log files and found there was a brief restart in the middle night during those days. Talked with the server administrator, and that may be the problem for the software to misread the structure information. This may give wrong frequencies that change the zero point energies. It's our fault to ignore the server problem. We truly appreciate the expertness and persistence of the reviewer to improve our manuscript.

Figure 6 Theoretical calculations. (a) Model of “Ni₂-S₁” active region in S-NiBDC. Free-energy diagrams for (b) water adsorption and (c) H adsorption of NiBDC and S-NiBDC. (d) PDOS of different sites for S-NiBDC with H binding. (e) Schematic illustration of the HER mechanisms through Volmer-Heyrovsky pathway on S-NiBDC. Color legend for atoms: gray, C; white, H; blue, Ni; red, O; yellow, S.

Corresponding revisions:

In response to this comment, we updated **Figure 6b-c** and revised the corresponding part on **Page 12 and Page 16** of the revised manuscript as follows:

1) The results and discussion part

Figure 6c shows the calculated adsorption Gibbs free energies of H* (ΔG_{H^*}), the value of which should be close to zero for ideal HER catalyst⁵². Compared with the NiBDC (-0.38 eV) as well as the Ni site (-0.31 eV) in S-NiBDC, the S site in S-NiBDC delivers the optimal value of ΔG_{H^*} (-0.12 eV) that is closer to the optimal value, revealing that the S site in the active region is most effective for H₂ evolution.

2) The calculation method part

A $3 \times 3 \times 3$ Monkhorst-Pack *k*-points grid was used for *k*-points sampling, and 900 eV plane-wave expansion was setup for energy cut-offs.

Comment 2: In Figure 3f, the stability test of NiBDC should also be conducted in a refreshed electrolyte similar to the S-NiBDC for a better comparison.

Authors' Response:

Thanks! We agree well with your comments that the stability test of NiBDC should be conducted in a refreshed electrolyte, which can display a better comparison with S-NiBDC sample. Following the reviewer's kind suggestion, we re-measured the HER stability at 1.0 A cm⁻² for NiBDC sample with electrolyte refreshed. As shown in **Figure 3g**, the HER catalytic activity for NiBDC still decreases rapidly with electrolyte refreshed, which is consistent with the conclusion drawn from the previous test.

Figure 3. (g) Chronopotentiometric curves at 1.0 A cm⁻² for NiBDC and S-NiBDC.

Corresponding revisions:

The results and discussion part

In response to this comment, we updated **Figure 3g** in the revised manuscript.

Comment 3: *In line 220 page 12, it is strange to demonstrate “the vibration absorption band of S-H* intermediate was not found for NiBDC”, because there is even no S element in the NiBDC sample.*

Authors' Response:

We appreciate the valuable comment from the reviewer. Following the reviewer's suggestion, we have changed the expression in the revised manuscript of *in-situ* ATR-FTIR tests for NiBDC sample.

Corresponding revisions:

The results and discussion part

In response to this comment, we revised the corresponding part on **Page 10** of the revised manuscript as follows:

In contrast, there was no significant intermediate peak observed for NiBDC in the same potential window (Figure 5d).

Thanks again for all your valuable comments to help us improve the quality of our manuscript!

Response to the comments of Reviewer #4

Comments: *This revised manuscript proposed S-doped NiBDC nanosheet arrays with the “Ni₂-S₁” active centers through a universal ligand regulation strategy. The electrocatalyst S-NiBDC was capable of reaching the current density of 1.0 A cm⁻² and maintaining stability for more than 150 h, which can be further increased to 3.5 A cm⁻². This electrocatalytic performance of S-NiBDC presents a breakthrough in noble-metal-free MOF-based HER electrocatalysts and makes a significant contribution to the research in this field. Detailed characterizations demonstrated the successful construction of the “Ni₂-S₁” mimic-hydrogenase active regions in the S-NiBDC structure. Moreover, the Operando EIS measurements, KIE tests, in-situ ATR-FTIR spectra, and DFT calculations confirmed the accelerated water activation kinetics and deeply revealed the electrocatalytic mechanism of HER at the constructed triangular “Ni₂-S₁” sites. The superior stability of S-NiBDC with a ternary active site compared to NiBDC was also verified.*

After carefully reading it, I find this work very interesting and stand high priority compared to recently reported MOF-based water splitting catalysts. The whole concept is well-established, the experiments are also very well done, the findings show good novelty, and the mechanism and in-situ studies provide an insightful understanding of alkaline HER on the Ni-based MOF. More importantly, the authors have responded well and thoroughly to the previous reviewers' comments; I think all the queries have been adequately addressed and well explained. Therefore, I would like to suggest that this significantly revised manuscript can be completely recommended for publication in Nature Communications. But before the full acceptance, I would like to suggest the authors further polish the manuscript via the following minor amendments.

Authors' Response:

Thank you very much for your positive and helpful comments on our manuscript. According to your comments, we have revised the manuscript with great attention.

Comment 1: *Figure 3f shows that the performance of NiBDC decreases rapidly during HER; what are the reasons for its performance degradation? Some explanation is needed.*

Authors' Response:

We appreciate the valuable comment from the reviewer. The results of *in-situ* electrochemical-Raman, post-XRD, and FESEM analyses on the NiBDC catalyst revealed the significant collapse of the BDC-based MOF structure (**Figure 4b, S25, S27**). Moreover, in **Figure S26**, the ¹H-NMR spectra of the electrolyte after HER reaction for NiBDC catalyst showed the obvious release of BDC organic ligands during the alkaline HER process, resulting from the damage of morphology and structure. Based on these results, one can conclude that

the performance degradation of NiBDC catalyst can be due to the fracture of the Ni-O bond between the ligand and metal node (*Nat. Mater.* 2022, 21, 673-680; *Angew.Chem. Int. Ed.* 2020, 59,13101-13108).

Figure 4. (b) *In-situ* Raman spectra NiBDC at -0.3 V vs. RHE for different times.

Figure S25. XRD patterns of NiBDC before and after HER tests.

Figure S27. FESEM images of NiBDC after HER tests at different current densities.

Figure S26. ¹H-NMR spectra of the electrolytes after HER tests for NiBDC and S-NiBDC.

Comment 2: *The result of EIS in Figure S17 showed obvious differences among the samples; could you provide the specific values?*

Authors' Response:

We appreciate the valuable comment from the reviewer. Following the reviewer's suggestions, the specific values for EIS results have been provided, and corresponding results have been added into the Supporting Information. As shown in **Table S2**, the smaller R_{ct} value of 19.6 Ω was observed for the S-NiBDC, as compared with that of NiBDC (98.5 Ω), implying a fast charge transfer ability of S-NiBDC (*Adv. Energy Mater.* 2019, 1900390; *Nat Commun* 2021, 12, 1369).

Table S2. The EIS results of the catalysts in 1.0 M KOH solution.

Catalysts	R_s/Ω	R_{ct}/Ω
NiBDC	4.33	98.5
S-NiBDC	3.18	19.6

Corresponding revisions:

In response to this comment, we added **Table S2** in supporting information and revised the corresponding part on **Page 8** of the revised manuscript as follows:

The rapid charge transfer for S-NiBDC is also supported by electrochemical impedance spectroscopy (EIS), in which the R_{ct} value for S-NiBDC is smaller than that of NiBDC (Figure S17 and Table S2).

Comment 3: *Figure 6e, the schematic illustration of the accelerated water activation for HER in S-NiBDC (right) and NiBDC (left) is too cartoon; I suggest the authors try to draw a new one from the molecule or atom level rather than the bulk materials, which will make the scheme more insightful to the readers.*

Authors' Response:

We appreciate the valuable comment from the reviewer. Following the reviewer's suggestion, we re-draw the schematic illustration (**Figure 6e**) of the S-NiBDC for the HER process via Volmer-Heyrovsky pathway based on an atomic perspective, and hope to provide the readers with a clearer view of the conclusions that we elucidated.

Figure 6. (e) Schematic illustration of the HER mechanisms through Volmer-Heyrovsky pathway on S-NiBDC.

Corresponding revisions:

In response to this comment, we updated **Figure 6e** in the revised manuscript.

Thanks again for your valuable comments to help us improve the quality of our manuscript!

REVIEWERS' COMMENTS

Reviewer #1 (Remarks to the Author):

The authors have addressed the remaining comments from this reviewer.

Reviewer #4 (Remarks to the Author):

The authors have carefully responded all reviewers' concerns, I think this paper is in very high quality now, the claims are all solid and well-supported by the added experiments, therefore, I think it can be accepted for publication in Nature Communications now.